# INFLUENCE GUIDED SAMPLING FOR DOMAIN ADAPTATION OF TEXT RETRIEVERS

## ABSTRACT

General-purpose open-domain dense retrieval systems must usually be trained with a large, eclectic mix of corpora and search tasks. How should these diverse corpora and tasks be sampled for training? Conventional approaches are to sample them uniformly, or proportional to their instance population sizes, or depend on human-level expert supervision. It is well known that the training data sampling strategy can greatly impact model performance. However, how to find the optimal strategy has not been adequately studied in the context of embedding models. We propose **Inf-DDS**, a novel reinforcement learning–driven sampling framework that adaptively reweighs training datasets guided by influence-based reward signals and is much more lightweight w.r.t. to GPU consumption. Our technique iteratively refines the sampling policy, prioritizing sampling from datasets that maximize the model performance on a target development set. We evaluate the efficacy of our sampling strategy on a wide range of text retrieval tasks, demonstrating strong improvements in retrieval performance and better adaptation compared to existing gradient-based sampling methods, while also being $1.5\times - 4\times$ cheaper than them in terms of GPU compute needed. Our sampling strategy achieves a **5.03** absolute NDCG@10 improvement while training a multilingual *bge-m3-dense*[1] model and an absolute NDCG@10 improvement of **0.94** while training *sentence-transformers/all-MiniLM-L6-v2*[2], even when starting from an expert assigned weights on a large pool of training datasets.

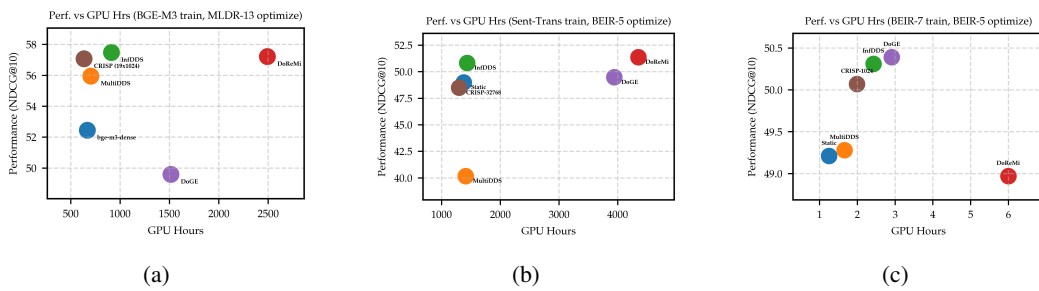

Figure 1: Training Time vs. Avg. NDCG@10: (a) BGE-M3 training optimized on MLDR-13 dev, (b) Sent-Trans training optimized on BEIR-5 dev, (c) BEIR-7 training optimized on BEIR-5 dev.

## 1 INTRODUCTION

Text-to-embedding based dense retriever models have recently gained huge popularity with their strong results across various benchmarks (Karpukhin et al., 2020; Reimers & Gurevych, 2019; Izacard et al., 2022; Gao & Callan, 2021; Wang et al., 2023; Xiao et al., 2022; Chen et al., 2024; BehnamGhader et al., 2024; Wang et al., 2022). These models are prized for their generalizability across domains without domain-specific tuning, typically trained on vast, diverse datasets. For instance, the Sentence-Transformer project[3], which develops generic sentence embeddings, utilized

---

[1] BAAI/bge-m3  [2] sentence-transformers/all-MiniLM-L6-v2  [3] huggingface.co/sentence-transformers

billions of instances across multiple datasets, with domain-specific data varying widely in size. However, larger datasets don't inherently improve embedding quality, making it crucial to identify the most informative datasets and their optimal proportions for training. Effective sampling strategies are essential to prevent overfitting or underfitting, making dataset selection a central challenge in developing robust, generalizable, or domain-specific embedding models.

While random sampling is a common default, it is limited by ignoring data source informativeness when sampling from large training datasets. Alternatives include temperature sampling and instance-based proportional sampling. A more intensive approach involves creating ad-hoc sampling distributions via iterative experimentation, termed expert weights, requiring expert evaluation. However, these strategies are static and predefined, often suboptimal compared to the unknown ideal distribution for maximizing model performance. A dynamic sampling approach, capable of adaptation, may better approximate this optimal distribution.

There has consequently been substantial interest in making sampling adaptive. Gradient-based approaches such as DDS (Wang et al., 2020a) and its multi-target extension (Wang et al., 2020b) use gradient-derived rewards to adjust the training distribution online. DoGE (Fan et al., 2024) proposes a *generalization estimation function* to approximate data influence, while methods like DoReMi (Xie et al., 2023a; Engstrom et al., 2024) rely on proxy models to estimate dataset utility. In practice, these dynamic methods face two main challenges: (1) instability and high variance introduced by stochastic gradients, which we empirically demonstrate in Section 4, and (2) substantial computational overhead when proxy models or expensive estimators are required. Together, these limitations motivate the design of an online, adaptive optimization strategy that can learn sampling weights efficiently and robustly while remaining computationally tractable.

In this paper, we propose **Inf**luence-guided **D**ynamic **D**ata **S**ampling strategy **(Inf-DDS)**, a computationally efficient novel algorithm that addresses the critical challenge of data sampling for domain adaptation, overcoming limitations of existing gradient-based methods. Inf-DDS iteratively takes small gradient-update steps on each domain's data, monitoring the impact on the downstream metric. Domains demonstrating greater performance improvements are subsequently assigned higher rewards. This adaptive sampling strategy, inspired by recent influence-based methods (Koh & Liang, 2017; Bae et al., 2022; Fan et al., 2024; Yu et al., 2024; Xia et al., 2024) for sampling training data across multiple domains, focuses learning on the most informative subsets of data. Our algorithm offers three key advantages over prior work: (1) it eliminates the dependence on noisy gradient estimates for reward computation, (2) it efficiently reuses computations from updating the parameterized sampling distribution $\psi$ parameters to also update the model parameters $\theta$, making it much more computationally efficient and (3) it produces more reliable and interpretable sampling trajectories for better downstream gains.

Our contributions in this work are as follows:

**a.** We propose Inf-DDS, an influence-guided reinforcement learning approach that learns to adjust sampling probabilities across diverse training datasets, improving target-domain retrieval performance while being much more computationally efficient.

**b.** We validate Inf-DDS against robust benchmarks, including BEIR datasets Thakur et al. (2021), Sentence-Transformers *all-MiniLM-L6-v2* Reimers & Gurevych (2019) and MLDR Chen et al. (2024), demonstrating significant improvements while optimizing for a target domain.

## 2 RELATED WORK

Recent work on training models with large, diverse data pools has explored both simple heuristic sampling and more adaptive, learned reweighting schemes. A common practice is to sample languages uniformly or via a temperature-scaled distribution that interpolates between uniform and size-proportional sampling. For example in language pretraining, Cooldown (Li et al., 2024b) demonstrates improvements in multilingual training by oversampling high-resource languages during the initial phases of training, while shifting to uniform sampling across high- and low-resource languages toward the end to enhance generalization. DoReMi leverages the loss gap between proxy models and the target model to optimize domain sampling weights for train set generalization.

Early works on domain- or task-specific adaptation select relevant subsets of data using either cross-entropy differences or simple classifiers Moore & Lewis (2010); Brown et al. (2020). DSIR (Xie et al., 2023b) employs importance sampling by assigning weights to training instances based on their hashed features for the target task, which then guide data selection. In contrast, CRISP (Grangier et al., 2025b) clusters the training data and assigns importance weights to clusters based on the frequency of source/target instances that fall into each cluster.

Recent influence estimation approaches guide data selection by identifying and prioritizing examples that most impact a model's predictions or performance, ensuring the most influential data are included in training (Grosse et al., 2023; Nikdan et al., 2025; Zhou et al., 2025). Methods such as LESS (Xia et al., 2024) and Quad (Zhang et al., 2025) use first-order approximations of Influence estimates via Taylor expansions of the change in target loss after a gradient update to select relevant instances. In contrast, we focus on learning domain weights *online* during training. Prior work shows that data models can accurately predict the influence of training samples on held-out target examples (Ilyas et al., 2022; Engstrom et al., 2024). MATES (Yu et al., 2024) similarly uses a small proxy model to locally estimate oracle influence after a single step, but small proxies may provide limited accuracy. Building on these insights, we propose leveraging online proxy models to compute exact influence scores as signals for learning sampling weights.

DDS and DoGE go beyond fixed heuristics by learning a scorer network—via bi-level optimization of scorer and model parameters—to upweight examples whose gradients align with a held-out development set. Wang et al. (2020b) extend DDS to multiple targets (MultiDDS) by learning per-language scoring functions that optimize performance across development sets. However, in practice, gradient-based strategies suffer from significant variance in the reward signal. In contrast, our method builds on MultiDDS and DoGE by replacing noisy gradient-based rewards with online-computed influence scores, derived from updated model parameters. This simplifies and stabilizes training by adjusting dataset-level sampling weights and reusing intermediate computations.

Extensive research has explored domain adaptation and universal generalization; providing a comprehensive review is beyond the scope of this paper. We refer interested readers to (Grangier et al., 2025a; Choi et al., 2023; Chung et al., 2023; Cao et al., 2024; Pruthi et al., 2020; Park et al., 2023; Liu et al., 2025; Grosse et al., 2023) and related works for further insights.

## 3 INFLUENCE GUIDED DYNAMIC DATA SAMPLING

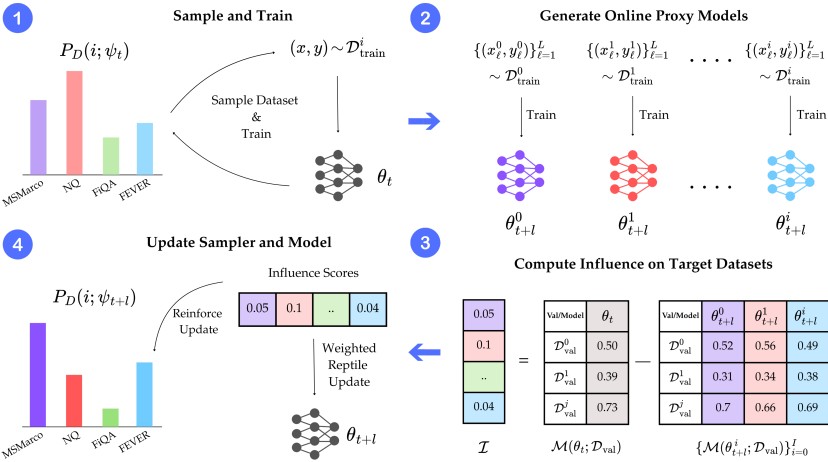

Figure 2: Overview of Inf-DDS. The trainable scorer $\psi$ and model parameters $\theta$ are optimized by generating online proxy models to compute influence scores, which serve as rewards for updating the scorer $\psi$. Proxy model gradients are efficiently reused for a weighted Reptile update on $\theta$.

In this section, we propose a reinforcement learning–based strategy for domain adaptation of text retrievers. We frame dataset sampling as a bilevel optimization problem and learn an adaptive policy to maximize performance on target datasets. Our approach is illustrated in Figure 2 and elaborated in Algorithm 1.

### 3.1 PROBLEM FORMULATION

**Efficient data sampling for training text retrieval models:** Our goal is to devise an adaptive sampling policy that efficiently samples batches from a large pool of training domains/datasets ($M$) to maximize the model performance on a set of target datasets ($N$) a.k.a. development or dev sets. We treat each training and dev dataset as a single homogeneous unit and optimize sampling at the dataset level.

Given a set of training datasets $\{\mathcal{D}_{\text{train}}^i\}_{i=1}^M$ with initial sampling probability as $P_D(i); i \in 1, \ldots, M$, a set of dev datasets $\{\mathcal{D}_{\text{dev}}^j\}_{j=1}^N$ on which we want our model to adapt, and initial model with parameters $\theta$, our objective is to optimize the model parameter $\theta$ by learning a dynamic sampling strategy for $P_D$ using a parameterized policy $\psi$ that maximizes performance on development sets.

We formalize our objective through the following optimization problem:

$$\theta^*, \psi^* = \arg\min_{\theta, \psi} \mathbb{E}_{i \sim P_D(i; \psi)} \left[ J(\theta, \mathcal{D}_{\text{train}}^i) \right] \tag{1}$$

where $J(\theta, \mathcal{D}_{\text{train}}^i)$ is the empirical risk.

The training datasets sampling probability distribution $P_D(i; \psi)$ is computed as:

$$P_D(i; \psi) = \frac{e^{\psi_i}}{\sum_{k=1}^M e^{\psi_k}} \tag{2}$$

where $M$ is the total number of datasets in the pool. This formulation allows the sampling strategy to dynamically adjust the probabilities based on the learned parameters ($\psi$), guiding the selection of datasets in a way that optimizes the model performance on the dev set. ($\psi_i$) represents the importance score associated with the training dataset $i$.

We assume that the dev sets $\mathcal{D}_{\text{dev}}$ have a distribution similar to the test set $\mathcal{D}_{\text{test}}$ that is, $\mathcal{D}_{\text{dev}} \approx \mathcal{D}_{\text{test}}$. Assuming the existence of $N$ development datasets, equation 1 can be expanded as follows:

$$\psi^* = \arg\min_{\psi} \frac{1}{N} \sum_{j=1}^N J(\theta^*(\psi), \mathcal{D}_{\text{dev}}^j) \; ; \quad \theta^* = \arg\min_{\theta} \mathbb{E}_{i \sim P_D(i; \psi)} \left[ J(\theta, \mathcal{D}_{\text{train}}^i) \right] \tag{3}$$

This formulation naturally leads to a bi-level optimization problem involving $\theta$ and $\psi$, which can be effectively addressed by alternating optimization steps. Specifically, the model parameters $\theta$ are updated using the standard gradient descent algorithm, while the scorer parameters $\psi$, are optimized using the REINFORCE algorithm (Williams, 1992).

### 3.2 INFLUENCE BASED REWARDS

Intuitively, the reward signal should guide the parameterized scoring network ($\psi$) to up-sample training datasets most likely to improve model performance on the development sets. We therefore propose an influence-based reward mechanism that quantifies the contribution of each training dataset to performance on a held-out development set. Existing influence-based methods (Xia et al., 2024; Fan et al., 2024; Yu et al., 2024) either rely on first-order approximations or simulate this contribution using smaller proxy models. In contrast, we employ online proxy models to obtain accurate influence scores. Specifically, at each time step $t$, we update the model parameters $\theta_t$ using dataset $\mathcal{D}_{\text{train}}^i$ for $l$ gradient steps, where $l$ is the minimum number of steps required to reach a meaningful local minima that demonstrates the potential benefit of up-sampling $\mathcal{D}_{\text{train}}^i$.

During each of these $l$ steps, we estimate the influence of every training subset $\mathcal{D}_{\text{train}}^i$ on the current model $\theta_t$. We do this by (1) taking $l$ gradient steps on $\mathcal{D}_{\text{train}}^i$ to produce $\theta_{t+1}^i$, (2) evaluating both $\theta_t$ and $\theta_{t+1}^i$ on each dev batch using an influence metric $\mathcal{M}$, and (3) computing the change in performance:

$$\Delta\mathcal{M}_j^i = \mathcal{M}(\theta_{t+1}^i; d_{\text{val}}^j) - \mathcal{M}(\theta_t^i; d_{\text{val}}^j) \propto \mathcal{I}_\theta(i; \mathcal{D}_{\text{val}})$$

where $\mathcal{I}_\theta(i; \theta)$ is the influence estimate on $\mathcal{D}_{\text{val}}$ if $\mathcal{D}_{\text{train}}^i$ is upweighed.

To ensure robustness and stability in our estimates, we normalize across the development datasets by taking a mean over all influences $\overline{\Delta\mathcal{M}^i}$ where $\overline{\Delta\mathcal{M}^i} = \frac{1}{N} \sum_{j=1}^N \Delta\mathcal{M}_j^i$. We iterate the process

$l$ times over all train–dev pairs $(i, j)$, accumulating the reward values $\overline{\Delta \mathcal{M}^i}$ to compute the final influence $\mathcal{I}^i$, which serves as a reliable measure of the impact of the $i^{th}$ training dataset on the model's performance on the development set.

---

**Algorithm 1** Pseudocode Inf-DDS

---

1: **Input:** $\{\mathcal{D}_{\text{train}}^i\}_{i=1}^M$, $\{\mathcal{D}_{\text{val}}^j\}_{j=1}^N$, influence metric $\mathcal{M}$, inner steps per meta-update $k$
2: **Output:** converged model $\theta^*$

3: Initialize $P_D(i; \psi, \tau) \leftarrow \frac{|\mathcal{D}_{\text{train}}^i|^{1/\tau}}{\sum_j |\mathcal{D}_{\text{train}}^j|^{1/\tau}}$
4: **while** $\theta_t$ not converged (every $k$ steps) **do**
5:      $\nabla_t \leftarrow 0$, $S \leftarrow 0$                 ▷ gradient cache
6:      Sample val batch $\{d_{\text{val}}^j\}_{j=1}^N \sim \mathcal{D}_{\text{val}}$
7:      **for** $i = 1, \ldots, M$ **do**                 ▷ parallel
8:          $(x, y) \sim \mathcal{D}_{\text{train}}^i$
9:          $\theta_{t+1}^i \leftarrow \text{Step}(\theta_t, \text{Opt}_t; x, y)$                 ▷ do this for $l$ steps
10:          $\Delta \mathcal{M}_j^i \leftarrow \mathcal{M}(\theta_{t+1}^i; d_{\text{val}}^j) - \mathcal{M}(\theta_t; d_{\text{val}}^j)$          ▷ compute influence
11:          $\mathcal{I}^i \leftarrow \frac{1}{N} \sum_j \Delta \mathcal{M}_j^i$
12:          $\nabla_t += \mathcal{I}^i(\theta_{t+1}^i - \theta_t), \quad S += \mathcal{I}^i$          ▷ accumulate influence updates
13:      **end for**
14:      $\bar{\nabla}_t \leftarrow \nabla_t / S$
15:      $\theta_{t+1} \leftarrow \theta_t + \alpha \bar{\nabla}_t$                 ▷ reward normalized reptile update
16:      $\text{Opt}_{t+1} \leftarrow \text{StateUpdate}(\text{Opt}_t, \bar{\nabla}_t)$
17:      $d_\psi \leftarrow \sum_{i=1}^M P_D(i; \psi) \, \mathcal{I}^i \, \nabla_\psi \log P_D(i; \psi)$
18:      $\psi \leftarrow \text{GradientUpdate}(\psi, d_\psi)$                 ▷ sampler update
19: **end while**

---

### 3.3 OPTIMIZATION FOR COMPUTE AND SCALABILITY

To reuse gradients across datasets and reduce computation, we perform Reptile-style first-order meta-updates (Nichol et al., 2018). For each training dataset $\mathcal{D}_{\text{train}}^i$ we take $l$ inner steps from the current initialization $\theta_t$ (step size $\eta_t$), producing $\theta_{t+1}^i$, and compute an influence score $\mathcal{I}^i$. We convert scores to a sampling distribution via softmax, $p_i = \exp(\mathcal{I}^i/\tau)/\sum_j \exp(\mathcal{I}^j/\tau)$, and form the weighted Reptile update

$$\bar{\theta}_{t+1} = \sum_{i=1}^M p_i \, \theta_{t+1}^i, \qquad \theta_{t+1} = \theta_t + \alpha(\bar{\theta}_{t+1} - \theta_t),$$

with Reptile rate $\alpha = \eta_t$. Using this procedure we need only a single copy of parameter gradients and optimizer states (first/second moments), which substantially reduces memory overheads.

When $M$ is large (e.g., many domains or languages) computing $I^i$ for every $i$ each iteration is costly. We therefore update the scorer $\psi$ on a uniform random subsample $S \subset \{\mathcal{D}_{\text{train}}^i\}_{i=1}^M$ with $|S| = k < M$. Restricting the policy to $S$ yields the conditional categorical

$$P_D(i \mid S; \psi) = \begin{cases} \dfrac{P_D(i; \psi)}{\sum_{j \in S} P_D(j; \psi)}, & i \in S, \\ 0, & i \notin S, \end{cases} \qquad P_D(i; \psi) = \frac{\exp(\psi_i)}{\sum_{j=1}^M \exp(\psi_j)}.$$

We then compute the scorer gradient on $S$ as

$$d_\psi = \sum_{i \in S} P_D(i; \psi) \, I^i \, \nabla_\psi \log P_D(i \mid S; \psi).$$

This estimator is unbiased for the *conditional* objective over $S$ (by the policy-gradient identity) but biased with respect to the full-objective gradient over all $M$ datasets. In practice, choosing $k < M$ yields large reductions in per-iteration compute and memory while still improving performance (see Figure 8 and Section 4).

# 4 EXPERIMENTAL SETUP

In this section, we detail the benchmarks and experimental setup used to test the domain adaptation of text retrievers under different sampling strategies.

**BEIR:** We start with a controlled setup where in-domain train, dev, and test retrieval datasets are available, though their sizes vary across domains. We train on seven BEIR-15 datasets: MSMarco, NQ, FEVER, FiQA, HotpotQA, SciFact, and NFCorpus, first optimizing on the FEVER dev set and then extending to other datasets with available dev and test sets, including Quora, FiQA, HotpotQA, and DBpedia. NFCorpus is excluded from the target datasets due to noise, as many level-1 relevant passages are in fact irrelevant. Our biencoder models are initialized with a pretrained *roberta-base* model (Liu et al., 2019), and trained for 2 epochs with InfoNCE loss (van den Oord et al., 2018) as influence metric $\mathcal{M}$ with cross-batch negatives. The scorer network is warmed up for 50 steps and updated every 50 training steps.

**Multilingual Long Document Retrieval (MLDR):** We next examine the realistic setting of Multilingual Long Document Retrieval, using the BGE-M3 corpus originally employed to train the *bge-m3-dense* model. Adaptation is performed on the MLDR-13 development set, with evaluation on its corresponding test set. The biencoder model is initialized from a 568M parameter bge-m3-unsupervised checkpoint[4]. We treat each language as a domain and sample proportionally across all datasets in a language. The scorer network is warmed up for 500 steps and updated every 250 training steps. We use the *m3-kd-distill* loss from BGE-M3, with cross-batch negatives and 8 hard negatives, as our influence metric.

**Sentence-Transformers Embedding Dataset:** Finally, we consider a more challenging scenario where train and test datasets span diverse domains, using the publicly available *sentence-transformers embedding dataset*[5], originally used to train the *all-MiniLM-L6-v2* model. The training corpus contains 1 billion parallel sentences drawn from 32 datasets. Its data configuration includes carefully tuned sampling weights, referred to as *Expert* initialization, designed to optimize performance. To align with our target domain, we excluded the Reddit comments dataset due to its large size and the CodeSearchNet dataset, as it is unrelated to code-focused tasks. Our biencoder models are initialized with the pretrained *MiniLM-L6-H384-uncased* model, and we optimize performance jointly across all BEIR-5 dev sets. The scorer network is warmed up for 500 steps and updated every 250 steps. Following the toy setting, we use the standard InfoNCE loss as the influence metric with cross-batch negatives. Additional hyperparameters are listed in Appendix A.

**Baselines:** We compare our sampling algorithm against four categories of baselines: (i) static sampling methods (Temperature, Cooldown), (ii) a universal generalization approach (DoReMi), (iii) gradient-based task-adaptive sampling methods (MultiDDS, DoGE), and (iv) a cluster-level, task-adaptive importance-sampling method (CRISP). For all baselines, we follow the training guidelines recommended in their respective papers, with hyperparameter details provided in Appendix A. For CRISP, we construct clusters in powers of $32^x$: $x \in 1, 2$ for BEIR-train, $x \in 1, 2, 3$ for Sentence-Transformers, and $x \in 1, 2$ per language for BGE-M3.

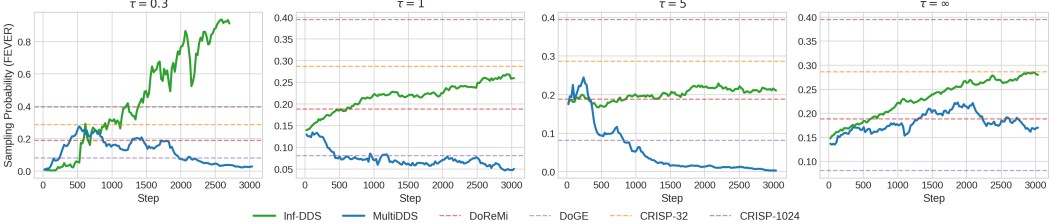

Figure 3: FEVER training set sampling trajectories for different initialization temperatures using MultiDDS, Inf-DDS, and learned weights from baseline methods (optimized on FEVER dev set).

---

[4] BAAI/bge-m3-unsupervised    [5] sentence-transformers/embedding-model-datasets

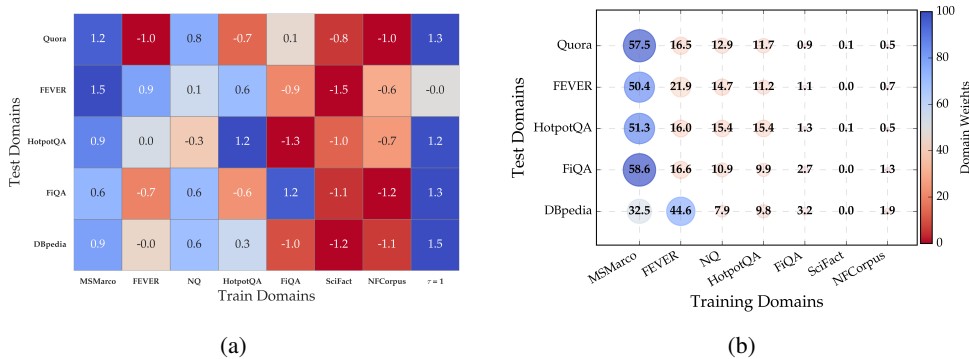

(a)                                    (b)

Figure 4: (a) Heatmap showing Z-score (row) normalized performance correlations between train and test splits across BEIR datasets. (b) Domain weights learned by Inf-DDS during optimization for each target domain.

# 5 RESULTS AND ANALYSIS

In this section, we aim to answer the following research questions: (1) Does learning a dynamically evolving sampling distribution through influence measures lead to superior adaptation on the test set? (2) Does the influence-based scorer capture additional insights beyond domain similarity between training and dev sets? (3) In a diverse domain setting, how reliable are influence-based approaches compared to gradient-based methods?

## 5.1 MAIN RESULTS

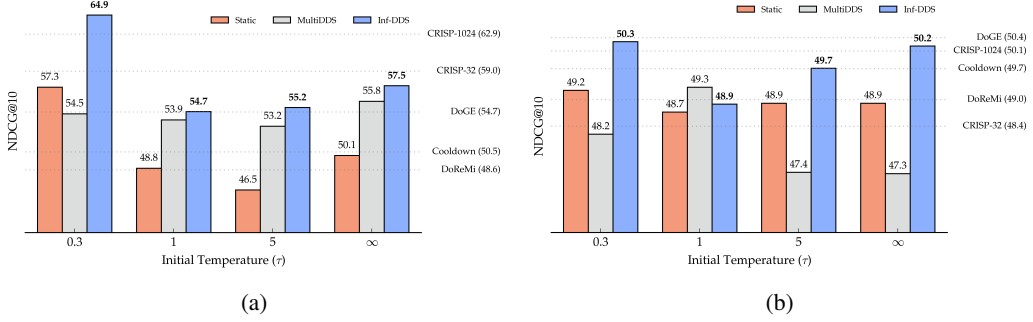

(a)                                    (b)

Figure 5: Sampling probability initialization vs Average NDCG@10 on the FEVER and BEIR-5 test set while training with BEIR-7 train set. (a) Scorer optimized jointly on the FEVER dev set (b) Scorer optimized jointly on BEIR-5 dev set.

**Domain adaptation on BEIR:** Our initial findings demonstrates that mere domain similarity alone does not consistently lead to enhanced performance in text retrieval setting. Figure 4a illustrates this by showing the normalized performance correlation between train/test sets without adaptive sampling. Notably, target datasets such as FEVER, HotpotQA, and FiQA benefit not only from their corresponding training domains but also from MS MARCO and $\tau = 1$ sampling, indicating that *effective retriever improvement requires going beyond domain similarity* highlighting the need for adaptive sampling strategies designed to optimize downstream performance.

To investigate this, we adapt a sampling distribution to the FEVER development set. As Inf-DDS relies on single-shot optimization, it is sensitive to the *initial distribution*, prompting us to assess multiple initializations to ensure robustness. As shown in Figure 5a, Inf-DDS consistently outperforms all baselines when initialized with $\tau = 0.3$, and remains competitive with CRISP using 32 clusters. Figure 3 provides a comparison of sampling trajectories, highlighting that MultiDDS exhibits unstable and inconsistent dynamics across varying temperatures, Inf-DDS produces stable behavior, consistently prioritizing FEVER and MS MARCO (Figure 4b), in line with CRISP

and DoReMi. This stability underscores the reliability of Inf-DDS in effectively aligning sampling strategies with measurable downstream performance improvements.

To further validate our approach, we conduct joint optimization over the BEIR-5 dev sets for generalization. As shown in Figure 5b, Inf-DDS outperforms static sampling in all temperature initializations and surpasses MultiDDS in 2 out of 3 scenarios. While MultiDDS shows more improvements when initialized with $\tau = 1$, it underperforms Inf-DDS' best score by $0.93$ points.

**MLDR:** Multilingual retrieval introduces distinct challenges stemming from the substantial variability in language resources and heterogeneous domain distributions. We assess how Inf-DDS and MultiDDS tackle these challenges by implicitly harmonizing data from high and low resource languages while leveraging cross-lingual relatedness without the need for explicit supervision. Specifically, we investigate two key questions: (1) Can Inf-DDS automatically upsample underrepresented languages within a shared multilingual corpus to improve performance? (2) How critical is sampling high-resource languages when optimizing for multiple languages?

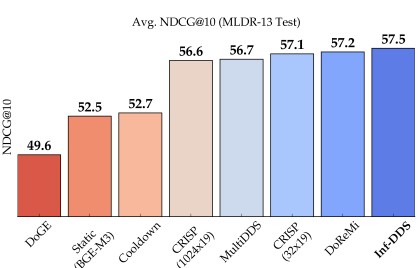

Figure 6: Avg. NDCG@10 scores on the MLDR-13 language test collection using BGE-M3 training data. Optimization of the scorer is done jointly on the 13 development sets.

We optimize the *bge-m3-unsupervised* model and scorer on the full MLDR-13 development set to maximize performance across 13 languages. As shown in Figure 6, starting from the same initial sampling weights as *bge-m3-dense*, Inf-DDS improves this baseline by $+5.03$ points in NDCG@10, while DoReMi achieves a $+4.76$-point gain. Inf-DDS also achieves the highest individual-language performance in 8 out of 13 languages.

Sampling trajectories for each language are presented in Figure 17 (Appendix). Interestingly, the sampling weights for English and Chinese drop substantially from their high initial values, yet performance remains comparable to *bge-m3-dense*, likely due to their dominant presence in *bge-m3-unsupervised* training (66.4% of total). This indicates that high-resource languages require little supervised data, demonstrating dynamic sampling's ability to upweight low-resource languages while avoiding overfitting on dominant ones. Notably, Swahili's sampling weight rises significantly despite minimal linguistic overlap with the development set; even with over half the training data sampled from Swahili, the model outperforms the baseline, though the cause of this effect remains unclear.

**Sentence-Transformers Embedding Dataset:** This diverse training corpus includes over 440 Million query-positive passage pairs spanning 32 domains. We use the same BEIR-5 development sets from the toy setting for optimization, as they have minimal overlap with the training data. This experiment addresses two key questions: (1) Does dynamic sampling remain effective with high domain diversity? (2) Can our algorithm further improve performance when a strong initial sampling distribution is available? The extensive domain coverage significantly increases the complexity of the adaptation problem.

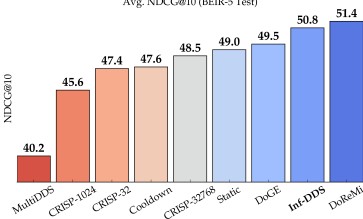 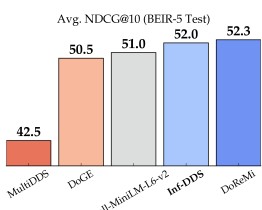

Figure 7: Average NDCG@10 on the BEIR-5 test collection using Sentence-Transformers training data with uniform (left) *expert* initialization (right). The scorer is optimized jointly on the development sets.

Starting from a uniform initialization, Inf-DDS achieves performance only $0.22$ points below the off-the-shelf Sentence-Transformers Expert model (*all-MiniLM-L6-v2*), nearly optimal—and yields

a 1.83 point gain over the uniform baseline. In contrast, gradient-based methods such as MultiDDS and DoGE fail to make any gains. When we re-run the experiment starting from Expert weights, Inf-DDS still produces an additional 0.94 point improvement, demonstrating its ability to refine even strong initial distributions. DoReMi attains a larger 1.25 point gain but requires $3\times$ the compute (Fig. 8). Figure 15 shows the evolving sampling distributions under Inf-DDS, illustrating that Expert weights can be further improved by dynamically adapting dataset sampling.

## 5.2 DISCUSSION

**Computational Overheads:** Inf-DDS computes exact influence scores for each training dataset using online proxy models. While this introduces additional overhead, retrieval models are typically small, making the tradeoff worthwhile given the performance gains. Figure 8 compares the training time of *all-MiniLM-L6-v2* across different sampling strategies. Although Inf-DDS is slightly slower than CRISP, MultiDDS, and static sampling, it consistently achieves superior performance. To reduce memory usage, Inf-DDS stores only a single set of intermediate gradients and optimizer states during influence computation, which are efficiently reused in the weighted Reptile update.

**Effect of initialization:** Choosing an effective initialization for sampling is challenging but can substantially impact Inf-DDS's performance (Figs. 5a, 5b and 7). While the algorithm does not always reach globally optimal sampling weights, starting from a reasonable initialization and updating the weights consistently yields gains. We do not explore heuristics for selecting initial weights, but experiments with standard static initializations show that, although no single choice is universally best, reasonably good initializations generally perform well.

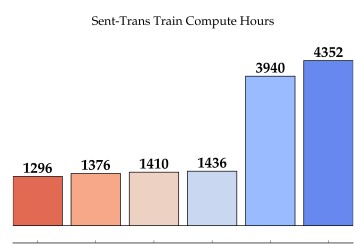

Figure 8: Comparison of approximate GPU Hours for training on Sent-Trans embedding data.

**Effect of Reptile Updates:** We perform an ablation on BEIR to assess the contribution of the meta-learning component in Table 11 (Appendix). Results show only minor differences when Reptile updates are disabled, indicating that most performance gains likely arise from dynamic sampling. Nevertheless, Reptile remains useful for reducing computational overhead by reusing existing computations.

**Relation between Influence and Gradients:** Stochastic gradient updates move model parameters toward the local minimum of the loss $\mathcal{L}$, whereas influence measures their impact on the target metric $\mathcal{M}$. When $\mathcal{L} \approx \mathcal{M}$, influence directly reflects the benefit of an optimization step. In contrast, gradient-based rewards quantify the alignment between the gradient toward the dev set minimum ($\nabla_{\text{dev}}$) and the gradient from a given training instance ($\nabla_{\text{train}}$). We posit that in high-dimensional landscapes, low alignment need not indicate convergence to a poor minimum. Influence-based rewards, by evaluating instances according to the actual target metric, provide a more direct and reliable estimate of which steps lead to the best attainable minima.

## 6 CONCLUSION

In this work, we present a comprehensive analysis for adapting text retrievers to target domains using influence-guided dynamic data sampling (Inf-DDS). Our approach parametrizes the sampling distribution with scorer parameters $\psi$ and performs bi-level optimization, jointly updating both the model parameters $\theta$ and the scorer parameters $\psi$, using influence scores as rewards. Across multiple benchmarks and baselines spanning diverse domains, Inf-DDS produces more stable sampling trajectories and consistently comes close to or outperforms both proxy-model and gradient-based approaches, while remaining computationally efficient. Although the algorithm does not always converge to the global optimum, it reliably delivers substantial improvements from reasonable initializations. We further analyze why gradient-based signals can mislead optimization, demonstrating that influence-based rewards offer a more robust estimate of the best attainable minima. For future work, we aim to investigate improved initialization strategies and more sophisticated optimization techniques for the parameterized scorer distribution $\psi$.

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

# A  HYPERPARAMETERS

## A.1  BEIR

For our experiments, we initialize our bi-encoder with *roberta-base* model, we set the initial learning rate to 2e-5, and train with mixed-precision (FP16) enabled. We used an in-batch negative sampling strategy with a temperature of 0.02, normalizing representations before contrastive scoring. Both training and evaluation batch sizes were 256 examples on a single NVIDIA-A100 80GB GPU. For scorer updates, we also use a batch size of 256. We use a linear learning rate decay with a warmup of 250 steps, and trained for a total of 7,000 steps. We use the standard InfoNCE van den Oord et al. (2018) as our loss and our influence metric $\mathcal{M}$. Queries and passages were truncated to maximum lengths of 64 and 256 tokens, respectively, with CLS-token pooling for sentence embeddings. We further update our scorer every 50 steps after a 50-step scorer warmup (we don't train the scorer during the warmup period). For reptile updates, we use $\alpha$ based on the current learning rate $\eta_t$ of the learning rate schedule. We do checkpoint selection by selecting the best checkpoint on the dev set and report the corresponding numbers on the final test set. For DoReMi, we train our reference models using $\tau = 1$ sampling and proxy models starting with $\tau = \infty$ initialization. For DoGE, too, we train our proxy models starting with $\tau = \infty$ initialization. For CRISP, we limit the clusters to $32^x$: $x \in 1, 2$ since the size of the total dataset is only 800k instances and use *all-MiniLM-L6-v2* embedding model for clustering.

## A.2  SENTENCE TRANSFORMERS EMBEDDING

We initialize our bi-encoder with *nreimers/MiniLM-L6-H384-uncased* model, we set the initial learning rate to 2e-5, and train with mixed-precision (FP16) enabled. We used an in-batch and cross-device negative sampling strategy with a temperature of 0.02, normalizing representations before contrastive scoring. Training batch sizes are set to 2000, and evaluation batch sizes are 256 examples per GPU and use 8 NVIDIA-A100 80GB GPUs. We use the standard InfoNCE van den Oord et al. (2018) as our loss and our influence metric $\mathcal{M}$. We also use 1-hard negative when using MSMarco in the training set. For scorer updates, we use a batch size of 256 per dataset. We use a linear learning rate decay with a warmup of 1000 steps, and trained for a total of 150k steps. Queries and passages were truncated to maximum lengths of 64 and 256 tokens, respectively, with CLS-token pooling for sentence embeddings. We further update our scorer every 250 steps after a 500-step scorer warmup (we don't train the scorer during the warmup period). For reptile updates, we use $\alpha$ based on the current learning rate $\eta_t$ of the learning rate schedule. For DoReMi, we train our reference models using $\tau = \infty$ sampling when comparing for uniform initialization $\tau =$ expert for expert initialization and proxy models starting with $\tau = \infty$ initialization. For DoGE, too, we train our proxy models starting with $\tau = \infty$ initialization sampling when comparing for uniform initialization $\tau =$ expert for expert initialization. For CRISP, we limit the clusters to $32^x$: $x \in 1, 2, 3$ since the size of the total dataset is only 440M instances, and use *all-MiniLM-L6-v2* embedding model for clustering.

## A.3  MULTILINGUAL LONG DOCUMENT RETRIEVAL

We initialize our bi-encoder with *BAAI/bge-m3-unsupervised* model, we set the initial learning rate to 2e-5, and train with mixed-precision (FP16) enabled. We used an in-batch negative, cross-device negatives, and 8 hard negatives during training with a temperature of 0.02, normalizing representations before contrastive scoring. We use the *bge-m3-kd-distil* loss by BGE-M3 as our training loss and InfoNCE with hard negatives as our influence metric $\mathcal{M}$. Training batch sizes are set to 5, and evaluation batch sizes are 5 examples per GPU and use 8 NVIDIA-A100 80GB GPUs. For scorer updates, we use a batch size of 4 per dataset. We use a linear learning rate decay with a warmup of 1000 steps, and trained for a total of 10k steps. Queries and passages were truncated to maximum lengths of 512 and 8192 tokens, respectively, with CLS-token pooling for sentence embeddings. We further update our scorer every 250 steps after a 500-step scorer warmup (we don't train the scorer during the warmup period). We employ subsampling as detailed in Section 3.3 with $k = 8$. We experiment with both enabling and disabling Reptile updates, and observe better performance when Reptile updates are turned off in the MLDR-13 setting. We use reptile update $\alpha$ based on the current learning rate $\eta_t$ of the learning rate schedule. For DoReMi, we train our reference models using $\tau = \infty$ sampling and proxy models starting with $\tau = \infty$ initialization. For DoGE, too,

we train our proxy models starting with $\tau = \infty$ initialization. For CRISP, we limit the clusters to $32^x$: $x \in 1, 2$ per language since the size of the total dataset is only 1.5M instances and use *paraphrase-multilingual-MiniLM-L12-v2* embedding model for clustering.

# B  DATASETS

## B.1  BEIR

Table 1: BEIR Train Datasets

| Dataset | #Qrels |
|---------|--------|
| MSMARCO | 499,184 |
| NFCorpus | 2,590 |
| NQ | 100,231 |
| HotpotQA | 85,000 |
| FiQA | 5,500 |
| Fever | 109,810 |
| SciFact | 809 |
| **Total** | **803,124** |

Table 2: BEIR Dev Datasets

| Dataset | #Qrels |
|---------|--------|
| MSMARCO | 7,437 |
| HotpotQA | 10,894 |
| FiQA | 1,238 |
| Quora | 7,626 |
| DBpedia | 5,673 |
| Fever | 8,079 |
| **Total** | **40,947** |

## B.2  SENTENCE TRANSFORMERS EMBEDDING DATASET

The all-MiniLM-L6-v2 model was fine-tuned with a self-supervised contrastive objective over a concatenation of diverse, publicly available sentence-pair corpora. These include user-generated content such as paired Reddit comments and Q&A threads from Stack Exchange and Yahoo Answers; scientific citation pairs drawn from the S2ORC and SPECTER datasets; question answering benchmarks like PAQ, MSMARCO, Natural-Questions, SearchQA, SQuAD 2.0, and TriviaQA; paraphrase and duplicate-question collections from WikiAnswers, Quora Question Triplets, and the AllNLI (SNLI+MultiNLI) corpus; multimodal captions from COCO and Flickr30k; code-search examples; and specialized text-compression and instructional corpora such as Simple Wikipedia, Wikihow, Altlex, and explicit sentence-compression datasets.

## B.3  BGE-M3 MULTILINGUAL

For English, bge-m3 fine-tuning dataset includes 8 datasets, including HotpotQA Yang et al. (2018), TriviaQA Joshi et al. (2017), NQ Kwiatkowski et al. (2019), MS MARCO Nguyen et al. (2016), COLIEE Kim et al. (2022), PubMedQA Jin et al. (2019), SQuAD Rajpurkar et al. (2016), and SimCSE Gao et al. (2021). For Chinese, it includes 7 datasets, DuReader He et al. (2018), mMARCO-ZH Bonifacio et al. (2021), T2-Ranking Xie et al. (2023c), LawGPT Zhou et al. (2024), CMedQAv2 Zhang et al. (2018), NLIzh2, and LeCaRDv2 Li et al. (2024a). It also includes training data for other languages from Mr. Tydi Zhang et al. (2021), MIRACL Zhang et al. (2023) and train sets of MLDR.

# C  ADDITIONAL RESULTS

## C.1  BEIR

To further validate our approach, we conduct individual optimizations on each of the BEIR-5 development sets and report the results in Table 7 Figure 9. As shown, Inf-DDS consistently matches or outperforms static sampling across all datasets. While it does not achieve the highest performance in every domain compared to MultiDDS, Inf-DDS surpasses proportional sampling by an average margin of 2.24 points and even outperforms MultiDDS on average. The corresponding sampling trajectories are provided in the Appendix D.

| Dataset | Count | Proportional Sampling % | Weight | Expert Sampling % |
|---|---|---|---|---|
| S2ORC Citation pairs (Abstracts) | 116,288,806 | 26.42% | 123 | 3.07% |
| WikiAnswers Duplicate question pairs | 77,427,422 | 17.60% | 123 | 3.07% |
| Amazon QA Pairs | 2,448,839 | 0.56% | 247 | 6.16% |
| PAQ (Question, Answer) pairs | 64,371,441 | 14.63% | 123 | 3.07% |
| S2ORC Citation pairs (Titles) | 52,603,982 | 11.96% | 123 | 3.07% |
| S2ORC (Title, Abstract) | 41,769,185 | 9.49% | 123 | 3.07% |
| Stack Exchange (Title, Body) pairs | 25,316,456 | 5.75% | 565 | 14.09% |
| Stack Exchange (Title+Body, Answer) pairs | 21,396,559 | 4.86% | 17 | 0.42% |
| Stack Exchange (Title, Answer) pairs | 21,396,559 | 4.86% | 373 | 9.31% |
| Stack Exchange Math | 2,218,989 | 0.50% | 166 | 4.14% |
| MS MARCO triplets | 9,144,553 | 2.08% | 247 | 6.16% |
| GOOAQ: Open QA with Diverse Answer Types | 3,012,496 | 0.68% | 247 | 6.16% |
| Yahoo Answers (Title, Answer) | 1,198,260 | 0.27% | 247 | 6.16% |
| COCO Image captions | 828,395 | 0.19% | 1 | 0.02% |
| SPECTER citation triplets | 684,100 | 0.16% | 84 | 2.10% |
| Yahoo Answers (Question, Answer) | 681,164 | 0.15% | 169 | 4.21% |
| Yahoo Answers (Title, Question) | 659,896 | 0.15% | 163 | 4.07% |
| SearchQA | 582,261 | 0.13% | 144 | 3.59% |
| Eli5 | 325,475 | 0.07% | 81 | 2.02% |
| Flickr 30k | 317,695 | 0.07% | 1 | 0.02% |
| Stack Exchange Duplicate questions (titles) | 304,525 | 0.07% | 26 | 0.65% |
| AllNLI (SNLI and MultiNLI) | 277,230 | 0.06% | 69 | 1.72% |
| Stack Exchange Duplicate questions (bodies) | 250,519 | 0.06% | 21 | 0.52% |
| Stack Exchange Duplicate questions (titles+bodies) | 250,460 | 0.06% | 21 | 0.52% |
| Sentence Compression | 180,000 | 0.04% | 45 | 1.12% |
| Wikihow | 128,542 | 0.03% | 32 | 0.80% |
| Altlex | 112,696 | 0.03% | 28 | 0.70% |
| Quora Question Triplets | 103,663 | 0.02% | 26 | 0.65% |
| Simple Wikipedia | 102,225 | 0.02% | 26 | 0.65% |
| Natural Questions (NQ) | 100,231 | 0.02% | 25 | 0.62% |
| SQuAD2.0 | 87,599 | 0.02% | 22 | 0.55% |
| TriviaQA | 73,346 | 0.02% | 19 | 0.47% |
| **Total** | **440,096,511** | **100.00%** | **4009** | **100.00%** |

Table 3: Training data provided by sentence transformers *all-MiniLM-L6-v2* showing dataset sizes, hand-picked weights, and normalized sampling percentages.

| Language | Sampling (%) |
|---|---|
| Swahili | 0.588 |
| Farsi | 0.588 |
| Finnish | 0.294 |
| Indonesian | 0.294 |
| French | 1.176 |
| German | 1.176 |
| Korean | 1.176 |
| Spanish | 1.176 |
| Italian | 1.176 |
| Portuguese | 1.176 |
| Japanese | 1.176 |
| Bengali | 0.294 |
| Telugu | 0.294 |
| Thai | 1.176 |
| Russian | 2.353 |
| Hindi | 1.176 |
| Arabic | 2.353 |
| Chinese | 23.529 |
| English | 58.824 |

Table 4: Language wise initialization probabilities for *bge-m3-dense* training.

## C.2 SENTENCE TRANSFORMERS EMBEDDING DATASET

Here in Table 7 we report Average NDCG@10 numbers on BEIR-5 test collection when training *all-MiniLM-L6-v2* using Sentence-Transformers data, when optimization of scorer is done jointly on the BEIR-5 development sets.

## C.3 MLDR

Here in Table 10 we report Average NDCG@10 numbers on MLDR-13 test set when training with BGE-M3 data, when optimization of scorer is done jointly on the MLDR-13 development sets.

| Dataset | Lines | Sampling % |
|---|---|---|
| MSMarco | 485,905 | 30.95% |
| MIRACL/fr | 1,143 | 0.07% |
| MIRACL/zh | 1,312 | 0.08% |
| MIRACL/es | 2,162 | 0.14% |
| MIRACL/ja | 3,477 | 0.22% |
| MIRACL/te | 3,452 | 0.22% |
| MIRACL/en | 2,863 | 0.18% |
| MIRACL/id | 4,071 | 0.26% |
| MIRACL/fa | 2,107 | 0.13% |
| MIRACL/ko | 868 | 0.06% |
| MIRACL/fi | 2,897 | 0.18% |
| MIRACL/th | 2,972 | 0.19% |
| MIRACL/bn | 1,631 | 0.10% |
| MIRACL/ru | 4,683 | 0.30% |
| MIRACL/hi | 1,169 | 0.07% |
| MIRACL/ar | 3,495 | 0.22% |
| MIRACL/sw | 1,901 | 0.12% |
| HotpotQA | 84,516 | 5.38% |
| mMARCO-zh/chinese | 100,000 | 6.37% |
| NQ | 58,568 | 3.73% |
| zh_NLI/LCQMC | 10,000 | 0.64% |
| zh_NLI/BQ | 12,599 | 0.80% |
| zh_NLI/STS-B | 249 | 0.02% |
| zh_NLI/afqmc | 10,534 | 0.67% |
| zh_NLI/ATEC | 11,325 | 0.72% |
| zh_NLI/QBQTC_v2 | 10,000 | 0.64% |
| zh_NLI/PAWSX | 10,000 | 0.64% |
| DuReader | 80,416 | 5.12% |
| cMedQAv2 | 50,000 | 3.18% |
| TriviaQA | 60,315 | 3.84% |

| Dataset | Lines | Sampling % |
|---|---|---|
| Mr.TyDi/finnish | 6,561 | 0.42% |
| Mr.TyDi/bengali | 1,713 | 0.11% |
| Mr.TyDi/russian | 5,366 | 0.34% |
| Mr.TyDi/swahili | 2,072 | 0.13% |
| Mr.TyDi/indonesian | 4,902 | 0.31% |
| Mr.TyDi/arabic | 12,377 | 0.79% |
| Mr.TyDi/english | 3,547 | 0.23% |
| Mr.TyDi/korean | 1,295 | 0.08% |
| Mr.TyDi/japanese | 3,697 | 0.24% |
| Mr.TyDi/telugu | 3,880 | 0.25% |
| Mr.TyDi/thai | 3,319 | 0.21% |
| T2Ranking | 90,467 | 5.76% |
| en_NLI/nli_for_simcse | 274,951 | 17.51% |
| Law-Medical/colliee | 463 | 0.03% |
| Law-Medical/law_gpt | 500 | 0.03% |
| Law-Medical/lecardv2 | 591 | 0.04% |
| Law-Medical/pubmed_qa | 500 | 0.03% |
| MLDR/hi | 1,618 | 0.10% |
| MLDR/es | 2,254 | 0.14% |
| MLDR/ru | 1,864 | 0.12% |
| MLDR/de | 1,847 | 0.12% |
| MLDR/ja | 2,262 | 0.14% |
| MLDR/fr | 1,608 | 0.10% |
| MLDR/ar | 1,817 | 0.12% |
| MLDR/ko | 2,198 | 0.14% |
| MLDR/en | 10,000 | 0.64% |
| MLDR/zh | 10,000 | 0.64% |
| MLDR/pt | 1,845 | 0.12% |
| MLDR/it | 2,151 | 0.14% |
| MLDR/th | 1,970 | 0.13% |
| SQuAD | 87,599 | 5.58% |
| **Total** | **1,569,864** | **100.00%** |

Table 5: Training data provided by *bge-m3* showing dataset sizes and normalized sampling percentages.

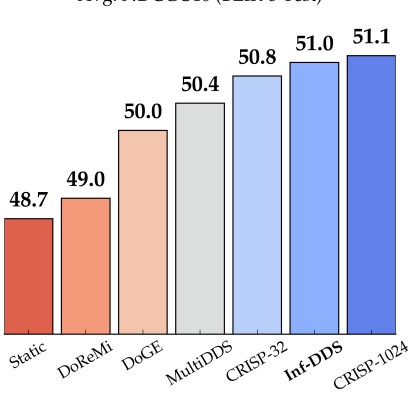

Avg. NDCG@10 (BEIR-5 Test)

Figure 9: Average NDCG@10 on the BEIR-5 test collection using BEIR-7 training data with $\tau = 1$. The scorer is optimized individually on the development sets.

## C.4 ABLATION

**Effect of reptile updates:** We perform an ablation study to determine whether the observed performance gains arise solely from dynamically updating the sampling distribution or whether the meta-learning component of our updates also contributes. Table 11 compares downstream performance with Reptile updates enabled versus disabled. We observe only minor differences in performance, which suggests that most of the improvement can be attributed to dynamic sampling. However, because the Reptile meta-update reuses existing computations, it remains valuable for reducing overall computational overhead.

| Init. | Static Sampling | MultiDDS | Inf-DDS | Others |
|---|---|---|---|---|
| $\tau = 0.3$ | 57.31 | 54.51 | 64.91 | - |
| $\tau = 1$ | 48.78 | 53.87 | 54.74 | - |
| $\tau = 5$ | 46.49 | 53.20 | 55.17 | - |
| $\tau = \infty$ | 50.12 | 55.83 | 57.47 | - |
| Cooldown | - | - | - | 50.5 |
| DoReMi | - | - | - | 48.6 |
| DoGE | - | - | - | 54.7 |
| CRISP-32 | - | - | - | 59.0 |
| CRISP-1024 | - | - | - | 62.9 |

Table 6: NDCG@10 comparison of sampling strategies with varying initialization temperatures during optimization on the FEVER dev set. Additional results from other baselines are shown in the last column.

| Sampling | Init. | Train Dataset | Dev Dataset | Average Test | DBpedia | FEVER | FiQA | HotpotQA | Quora |
|---|---|---|---|---|---|---|---|---|---|
| Static | $\tau = 1$ | BEIR-7 | - | 48.7 | **29.5** | 48.8 | 29.0 | 52.4 | **84.0** |
| MultiDDS | $\tau = 1$ | BEIR-7 | BEIR-1 | 50.4 | 27.4 | 53.9 | **33.3** | **53.8** | 83.7 |
| Inf-DDS | $\tau = 1$ | BEIR-7 | BEIR-1 | **51.0** | 29.4 | 54.7 | **33.7** | 52.9 | **84.1** |
| DoReMi | Proxy | BEIR-7 | - | 49.0 | 27.5 | 48.6 | 32.0 | 53.7 | 83.1 |
| DoGE | Proxy | BEIR-7 | BEIR-1 | 50.0 | 25.1 | 54.7 | 32.4 | **54.8** | 82.8 |
| CRISP-32 | Cluster IS | BEIR-7 | BEIR-1 | 50.8 | 27.0 | **59.0** | 31.3 | 52.9 | 83.6 |
| CRISP-1024 | Cluster IS | BEIR-7 | BEIR-1 | **51.1** | 24.8 | **62.9** | 31.8 | 52.6 | 83.3 |
| Static | $\tau = \infty$ | BEIR-7 | - | 48.9 | 27.2 | 50.1 | 31.7 | 52.8 | 82.8 |
| MultiDDS | $\tau = \infty$ | BEIR-7 | BEIR-1 | 49.8 | 25.6 | 55.8 | 31.1 | 53.1 | 83.1 |
| Inf-DDS | $\tau = \infty$ | BEIR-7 | BEIR-1 | 50.3 | 26.7 | 57.5 | 32.0 | 52.6 | 82.7 |

Table 7: Average NDCG@10 on BEIR-5 test collection, optimization of scorer is done individually on the development sets.

# D  SAMPLING TRAJECTORIES

# E  LIMITATIONS

Our experiments primarily focus on adapting text retrievers using a fixed set of development datasets. An important extension would be to explore how dynamically sampled training can help adapt already fine-tuned models to new domains. While our method shows consistent improvements, it does not always converge to the optimal sampling distribution. We observe that starting from a good initialization helps, but further work is needed to design strategies that can reach an optimal distribution with minimal iterations. Another important aspect is identifying heuristics to check whether dynamic sampling is applicable or not to a particular setting, since if the domains are conflated and it is not possible to reweigh domains. Additionally, our reported results are based on a single random seed. However, since our algorithm depends on the initial sampling distribution rather than random seed initialization, we observe stable improvements across benchmarks.

| Sampling | Init | Train Dataset | Dev Dataset | Avg. Test | Dbpedia | Fever | Fiqa | Hotpotqa | Quora |
|---|---|---|---|---|---|---|---|---|---|
| Static | $\tau = 0.3$ | | - | 49.21 | 28.64 | 58.21 | 26.53 | 48.74 | 83.97 |
| MultiDDS | | | BEIR-5 | 48.22 | 25.22 | 58.56 | 23.25 | 54.00 | 80.10 |
| Inf-DDS | | | | 50.31 | 27.73 | 69.24 | 23.64 | 48.63 | 82.31 |
| Static | $\tau = 1$ | BEIR-7 | - | 48.72 | 29.46 | 48.78 | 29.02 | 52.35 | 83.98 |
| MultiDDS | | | BEIR-5 | 49.28 | 25.93 | 58.99 | 28.98 | 50.54 | 81.95 |
| Inf-DDS | | | | 48.90 | 29.45 | 49.98 | 29.37 | 51.64 | 84.08 |
| Static | $\tau = 5$ | | - | 48.92 | 27.62 | 48.39 | 32.35 | 53.21 | 83.04 |
| MultiDDS | | | BEIR-5 | 47.36 | 24.40 | 51.04 | 30.13 | 49.74 | 81.49 |
| Inf-DDS | | | | 49.71 | 28.84 | 54.96 | 31.16 | 51.38 | 82.21 |
| Static | $\tau = \infty$ | | - | 48.92 | 27.24 | 50.12 | 31.71 | 52.79 | 82.76 |
| MultiDDS | | | BEIR-5 | 47.33 | 22.73 | 52.38 | 30.36 | 50.07 | 81.10 |
| Inf-DDS | | | | 50.21 | 26.77 | 59.58 | 32.09 | 50.44 | 82.16 |
| DoReMi | Proxy | BEIR-7 | - | 49.0 | 27.5 | 48.6 | 32.0 | 53.7 | 83.1 |
| DoGE | Proxy | BEIR-7 | BEIR-5 | 50.4 | 26.7 | 61.3 | 30.4 | 51.4 | 82.2 |
| CRISP-32 | Cluster IS | BEIR-7 | BEIR-5 | 50.0 | 26.2 | 57.2 | 31.0 | 52.3 | 83.5 |
| CRISP-1024 | Cluster IS | BEIR-7 | BEIR-5 | 49.4 | 23.6 | 56.1 | 31.2 | 52.5 | 83.3 |

Table 8: Average NDCG@10 on BEIR-5 test collection, optimization of scorer is done jointly on the development sets. Additional baselines are appended below.

| Sampling | Init. | Train Dataset | Dev Dataset | Average Test | DBpedia | FEVER | FiQA | HotpotQA | Quora |
|---|---|---|---|---|---|---|---|---|---|
| Static | $\tau = \infty$ | Sent-Trans | - | 48.99 | 31.71 | 48.98 | 31.82 | 44.59 | 48.98 |
| MultiDDS | $\tau = \infty$ | Sent-Trans | BEIR-5 | 40.17 | 22.27 | 43.94 | 24.76 | 26.49 | 83.87 |
| InfDDS | $\tau = \infty$ | Sent-Trans | BEIR-5 | 50.82 | 32.03 | 50.62 | 34.35 | 46.55 | 88.01 |
| Cooldown | $5 \to 1$ | Sent-Trans | - | 47.57 | 28.64 | 49.90 | 33.65 | 38.09 | 87.58 |
| DoReMi | Proxy | Sent-Trans | - | 51.36 | 31.71 | 52.31 | 36.17 | 49.09 | 87.59 |
| DoGE | Proxy | Sent-Trans | BEIR-5 | 49.47 | 29.90 | 57.39 | 31.15 | 42.61 | 86.32 |
| CRISP (n=32) | Cluster IS | Sent-Trans | BEIR-5 | 47.43 | 28.27 | 58.10 | 27.86 | 38.03 | 84.89 |
| CRISP (n=$32^2$) | Cluster IS | Sent-Trans | BEIR-5 | 45.64 | 26.79 | 54.79 | 26.39 | 35.52 | 84.73 |
| CRISP (n=$32^3$) | Cluster IS | Sent-Trans | BEIR-5 | 48.49 | 28.38 | 61.93 | 28.45 | 39.51 | 84.19 |
| Static (all-MiniLM-L6-v2) | Expert | Sent-Trans | - | 51.04 | 32.33 | 51.93 | 36.87 | 46.51 | 87.55 |
| MultiDDS | Expert | Sent-Trans | BEIR-5 | 42.47 | 22.79 | 37.21 | 29.24 | 28.25 | 86.25 |
| InfDDS | Expert | Sent-Trans | BEIR-5 | 51.98 | 32.35 | 58.31 | 36.14 | 45.71 | 87.38 |
| DoReMi | Expert | Sent-Trans | - | 52.29 | 32.54 | 56.32 | 36.22 | 48.68 | 87.67 |
| DoGE | Expert | Sent-Trans | BEIR-5 | 50.49 | 30.01 | 59.64 | 33.25 | 43.10 | 86.44 |

Table 9: Average NDCG@10 on BEIR-5 test collection when training *all-MiniLM-L6-v2* using Sentence-Transformers data, optimization of scorer is done jointly on BEIR-5 development sets.

| Model/ Sampling | Dev Dataset | Avg. Test | ar | de | en | es | fr | hi | it | ja | ko | pt | ru | th | zh |
|---|---|---|---|---|---|---|---|---|---|---|---|---|---|---|---|---|
| bge-m3 unsup. | - | 37.02 | 30.75 | 38.39 | 34.24 | 61.35 | 53.05 | 23.90 | 43.72 | 33.27 | 24.87 | 59.66 | 45.83 | 18.58 | 13.69 |
| bge-m3-dense | - | 52.45 | 47.60 | 46.10 | 48.90 | 74.80 | 73.80 | 40.70 | 62.70 | 50.90 | 42.90 | 74.40 | 59.50 | 33.60 | 26.00 |
| bge-m3-dense* | - | 52.45 | 48.41 | 46.66 | 46.70 | 76.30 | 74.28 | 39.98 | 61.69 | 49.14 | 41.00 | 74.25 | 60.75 | 36.03 | 26.68 |
| MultiDDS | MLDR-13 | 56.67 | 55.95 | 53.85 | 49.99 | 79.11 | 76.33 | 44.13 | 66.02 | 56.58 | 49.29 | 78.58 | 62.34 | 37.30 | 27.25 |
| Inf-DDS | MLDR-13 | 57.48 | 58.33 | 54.43 | 48.54 | 80.63 | 77.89 | 43.87 | 66.07 | 56.72 | 49.86 | 79.53 | 65.72 | 39.02 | 26.61 |
| Cooldown | - | 52.69 | 51.45 | 50.17 | 48.52 | 75.55 | 72.55 | 33.05 | 61.72 | 48.41 | 41.29 | 75.46 | 61.37 | 37.84 | 27.66 |
| DoReMi | - | 57.21 | 54.82 | 54.43 | 50.65 | 79.07 | 77.12 | 44.46 | 66.63 | 54.05 | 47.99 | 82.32 | 64.48 | 39.92 | 27.76 |
| DoGE (Iter 2) | MLDR-13 | 49.59 | 44.88 | 46.96 | 47.08 | 72.65 | 70.46 | 34.59 | 58.51 | 44.39 | 36.97 | 73.25 | 56.72 | 33.15 | 25.04 |
| CRISP (n=32/lang) | MLDR-13 | 57.07 | 56.68 | 53.34 | 52.56 | 79.48 | 75.57 | 43.96 | 66.62 | 55.37 | 48.84 | 79.27 | 64.69 | 36.72 | 28.86 |
| CRISP (n=$32^2$/lang) | MLDR-13 | 56.58 | 53.07 | 53.87 | 52.02 | 79.81 | 75.49 | 44.94 | 65.49 | 54.53 | 48.61 | 77.98 | 62.76 | 38.48 | 28.53 |

Table 10: Avg. NDCG@10 scores on the Multilingual Longform Document Retrieval dataset across 13 languages. * indicates reproduced numbers.

| Reptile Update | Train Dataset | Dev Dataset | Fever Test | Fiqa Test | HotpotQA Test |
|---|---|---|---|---|---|
| On | BEIR-7 | BEIR-1 | 54.74 | 33.70 | 52.85 |
| Off | | | 56.48 | 28.77 | 52.73 |

Table 11: Average NDCG@10 on FEVER, FiQA, and HotpotQA test collection, optimization of scorer is done individually on the development sets without reptile updates.

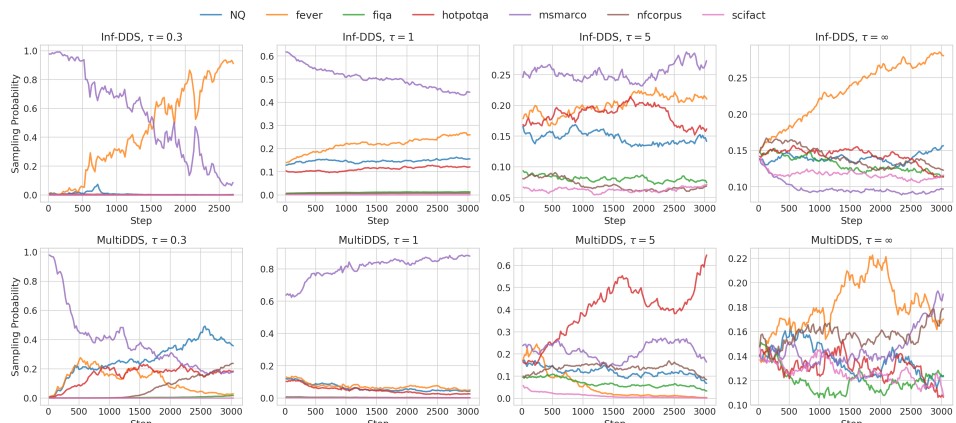

Figure 10: Sampling probability trajectories of MultiDDS and Inf-DDS with varying initialization temperatures during optimization on the FEVER development set. The orange curve denotes the FEVER training set sampling trajectory.

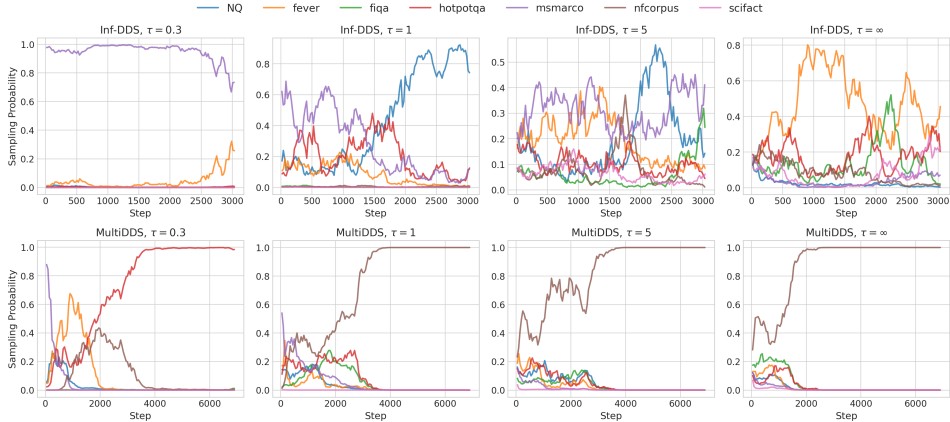

Figure 11: Sampling probability trajectories of MultiDDS and Inf-DDS with varying initialization temperatures during joint optimization on the BEIR-5 development set (DBpedia, FEVER, FiQA, HotpotQA, Quora). The brown curve represents the sampling trajectory for the NFCorpus training set, which is aggressively upsampled by MultiDDS, resulting in degraded overall performance, as shown in Table 8.

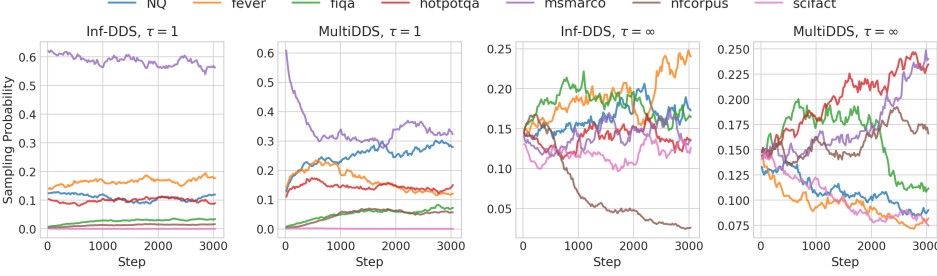

Figure 12: Sampling probability trajectories of MultiDDS and Inf-DDS with varying initialization temperatures during optimization on the FiQA development set. The green curve denotes the FiQA training set sampling trajectory. Compared to MultiDDS, which upsamples FiQA due to gradient similarity, Inf-DDS upsamples datasets like MSMarco and FEVER that are more relevant for performance gains as seen in Table 7.

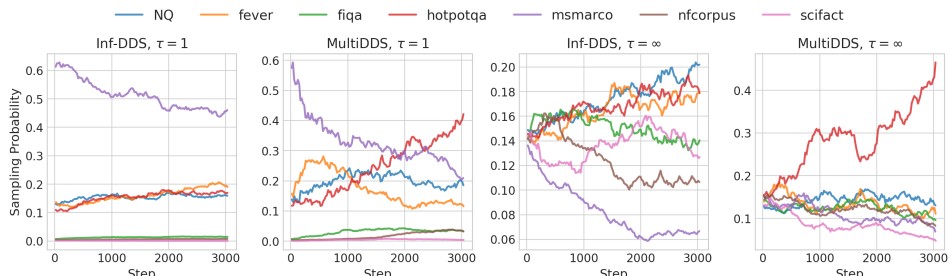

Figure 13: Sampling probability trajectories of MultiDDS and Inf-DDS with varying initialization temperatures during joint optimization on the HotpotQA development set. The red curve represents the sampling trajectory for the HotpotQA training set, which is being upsampled more by MultiDDS, leading to a better performance as seen in Table 7.

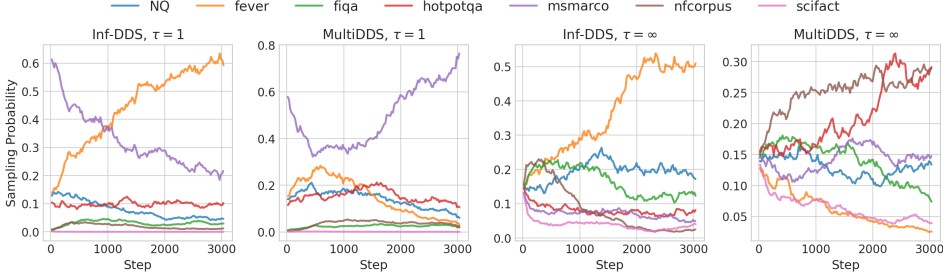

Figure 14: Sampling probability trajectories of MultiDDS and Inf-DDS with varying initialization temperatures during joint optimization on the Dbpedia development set. Since DBpedia is not present in the training set, we see FEVER being upsampled more by Inf-DDS, which should be true since DBpedia and FEVER are very closely related datasets Thakur et al. (2021).

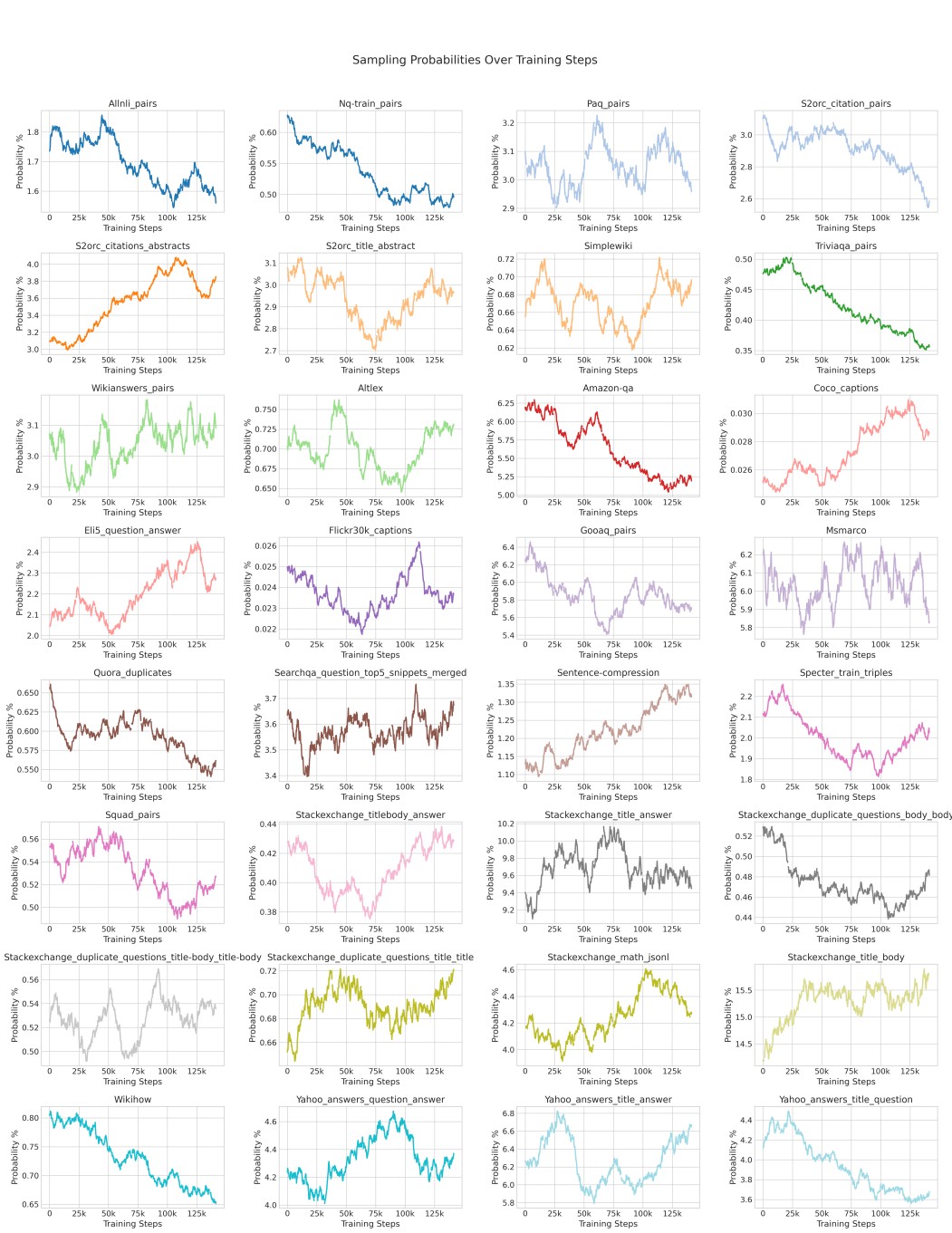

Figure 15: Sampling probability trajectories of Inf-DDS with *Expert* initialization during training of *all-MiniLM-L6-v2* on Sentence Transformers data. Optimization is done jointly on the BEIR-5 development set (DBpedia, FEVER, FiQA, HotpotQA, and Quora).

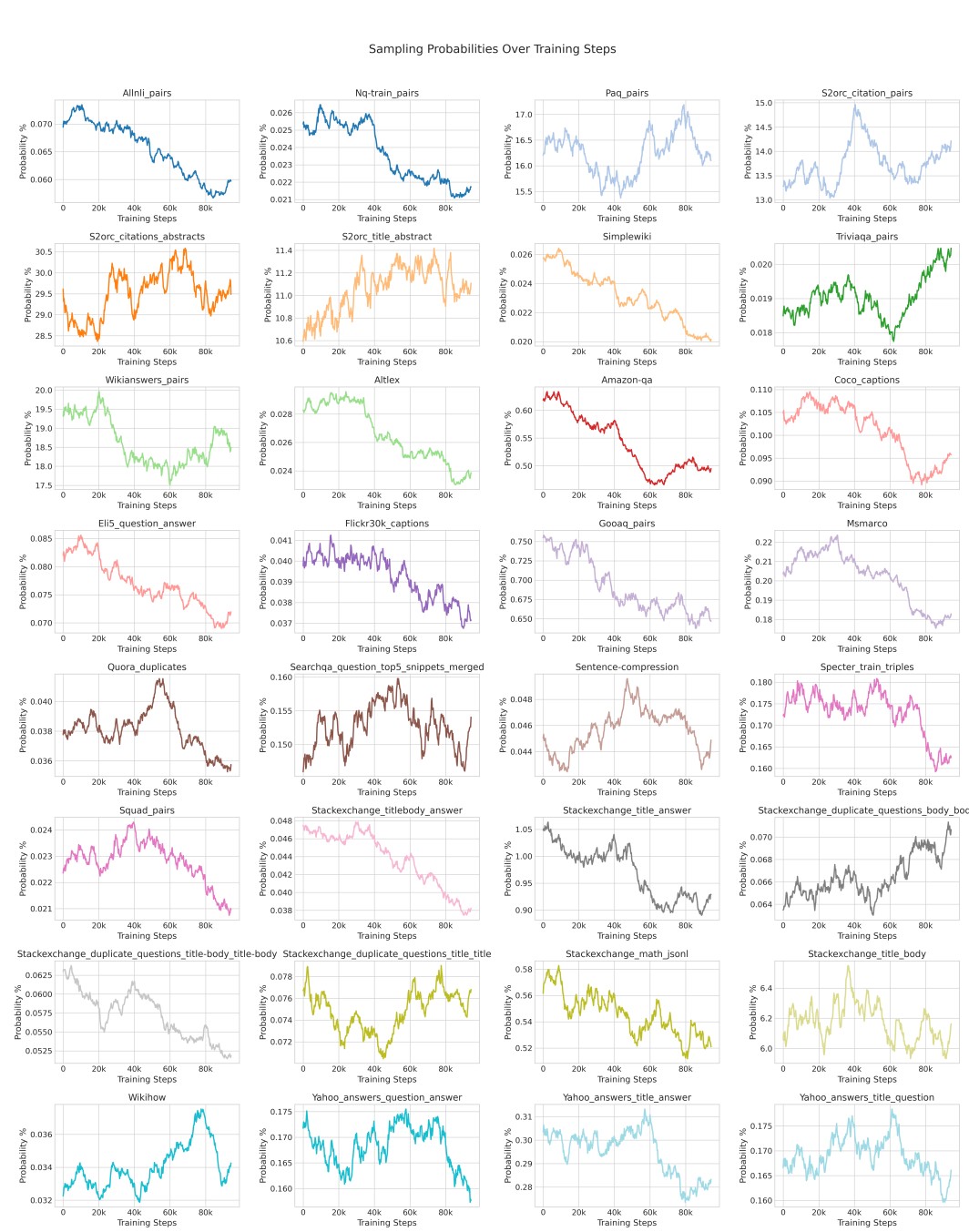

Figure 16: Sampling probability trajectories of Inf-DDS with $\tau = 1$ initialization during training of *all-MiniLM-L6-v2* on Sentence Transformers data. Optimization is done jointly on the BEIR-5 development set (DBpedia, FEVER, FiQA, HotpotQA, and Quora).

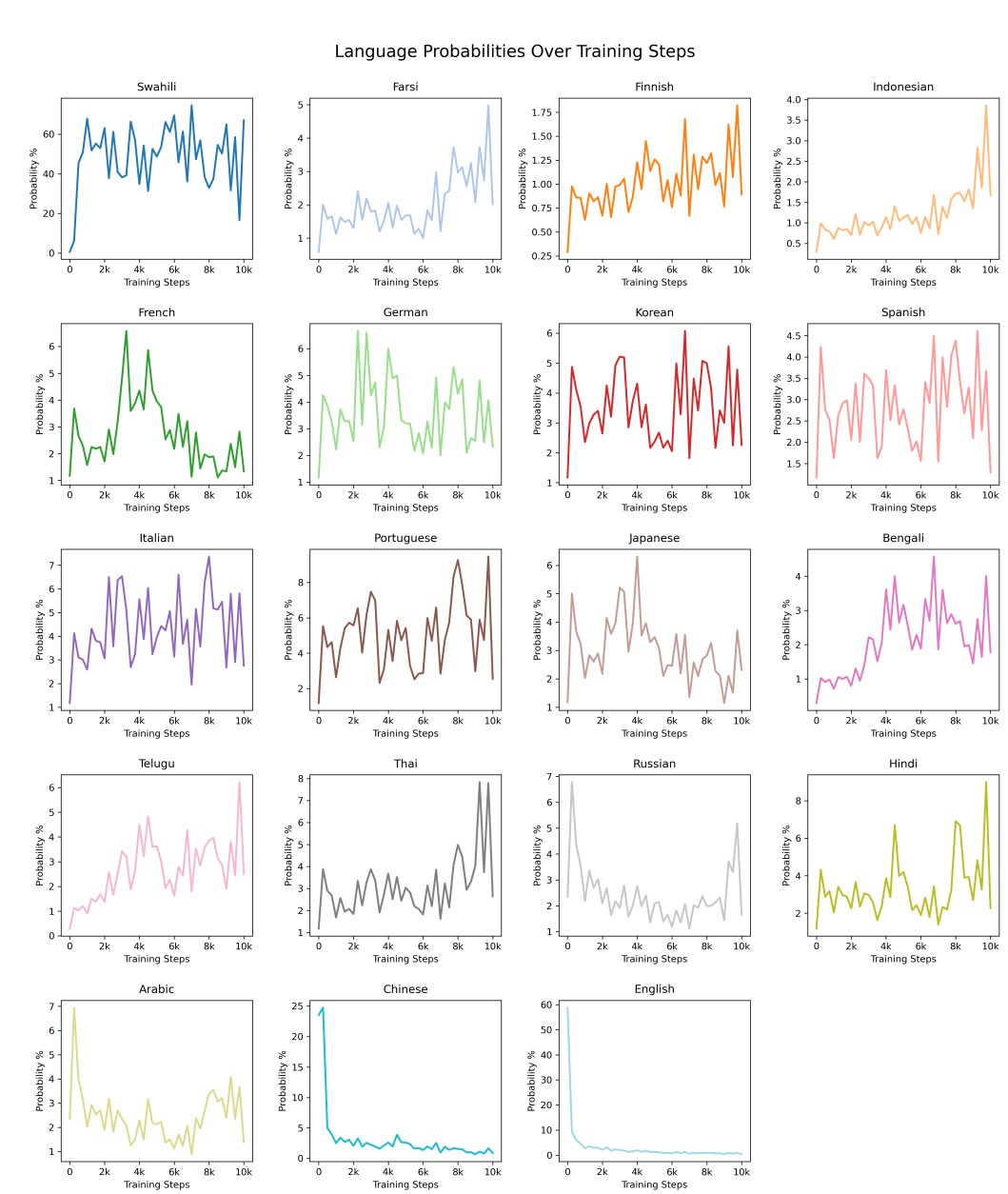

Figure 17: Sampling probability trajectories of Inf-DDS during training of *bge-m3-dense* while jointly optimizing for MLDR-13 dev sets.

