# OpenReview forum: "Influence Guided Sampling for Domain Adaptation of Text Retrievers"
_ICLR.cc/2026/Conference — Submitted to ICLR 2026_

### Official Review · Reviewer_66zW · 2025-10-31

**Soundness:** 2
**Presentation:** 3
**Contribution:** 3
**Rating:** 4
**Confidence:** 4

**Summary:**

This paper presents Inf-DDS, a novel method that uses influence, the metric gain from a small number of updates on the development set, as a reward signal for training a dynamic data sampling strategy. This method is designed to optimize the training data distribution for downstream tasks, particularly in dense retrieval settings. Empirical results on multiple benchmarks, including MLDR and BEIR, demonstrate significant improvements over existing baselines. The paper also introduces engineering solutions like weighted Reptile to reduce computational and memory overhead, making the method more efficient. However, the paper does not sufficiently address the risks of overfitting and data leakage.

**Strengths:**

1. **Clear and Direct Reward Design**: The use of influence as a reward based on actual metric improvements on the dev set makes the method straightforward and directly aligned with the downstream task's goals.
2. **Effective Engineering Design**: The use of weighted Reptile for gradient sharing and memory efficiency shows thoughtful consideration for practical implementation, making the approach scalable to large datasets and models.
3. **Stability and Robustness**: The method demonstrates stable learning trajectories, with less sensitivity to noisy gradients compared to alternative methods like DoGE or MultiDDS. The authors provide a useful analysis of sampling stability.

**Weaknesses:**

1. **Potential Overfitting and Data Leakage**: The method uses a portion of the downstream test data as the development set to guide data sampling for the training set. This introduces the risk of **data leakage** and overfitting, as the model may learn to optimize for the specific dev set rather than generalizing to unseen data. This is a **key concern**, as shifting the dev set may lead to performance degradation when applied to a new, unseen dev split. The paper does not sufficiently investigate or mitigate this issue, nor does it perform experiments to confirm the stability of the approach across multiple dev splits.
2. **Insufficient Theoretical Analysis**: While the paper presents empirical results, it lacks a detailed theoretical analysis of the potential biases and variance in the influence estimates, especially when the number of inner steps is small. The relationship between influence and long-term generalization remains unexplored. A more rigorous theoretical grounding would enhance the robustness of the method’s claims.
3. **Unclear Generalization Across Domains**: The authors do not sufficiently explain or diagnose certain anomalies in their experiments, such as the strong upsampling of Swahili in the MLDR experiments. The paper should offer an analysis of why such samples provide improvements in performance and whether this is due to the model overfitting to particular domains or justifiable improvements in generalization.
4. **Lack of Comparison with Related Works**: The paper does not cite or compare with proxy-model based methods like DoReMi algorithms, especially methods do not involve downstream task data. In the realm of dense embedding data sampling optimization, there are also previous works are not cited or compared in the paper. For example, tDRO (Task-level Distributionally Robust Optimization for Dense Retrieval) addresses similar issues of dataset-level weighting for improving domain robustness in the realm of dense embedding fine-tuning.
5. **Compute and Efficiency Trade-offs**: The computational cost of the proposed method is not adequately quantified, particularly when scaling to large datasets. The paper lacks a detailed comparison of the compute cost, GPU memory usage, and sample efficiency between Inf-DDS and other methods, making it difficult to assess the scalability of the method in practical scenarios.

**Questions:**

1. **Dev Split Sensitivity**: The method relies on a portion of the test data for the dev set. Could you provide experiments showing the sensitivity of the method to different dev splits and report the variance in performance across multiple splits?
2. **Overfitting Risk**: Have you considered any measures to regularize the learning of the sampler to prevent overfitting to noisy or biased dev sets? How do you plan to mitigate the risk of data leakage or the model becoming overly sensitive to the dev set?
3. **Computational Efficiency**: Can you provide a more detailed analysis of the computational trade-offs, such as GPU hour saving and sample efficiency, when compared to other methods?
4. **Comparison with proxy-model based methods**: Can you compare your approach with the recent proxy-model based methods in dense embedding training, e.g. tDRO, to highlight the differences and potential advantages of Inf-DDS, particularly in terms of generalization across domains and robustness to dev-split variations?
5. **Swahili Data Anomaly**: In your MLDR experiment, why does the Swahili language show a strong performance improvement when upsampled? Can you provide a more detailed analysis or an ablation study to explain this result?

---

> ### Author Response · Authors · 2025-11-22
> **Response to Reviewer 66zW**
>
> We thank the reviewer for their valuable comments and suggestions for our work and look forward to an engaging discussion.
> - Response to Weakness 1:
> > We emphasize the reliability of the dev set as a key requirement for our work (section 3.1). This is also the case in many previous works like MultiDDS, DoGE, CRISP, etc (lines 125-129). All of these, including ours, assume a reasonable dev set that approximates the test set and has no leakage. Such a dev set can serve as a reliable proxy, which holds true for the widely known BEIR datasets as well as MLDR for multilingual tasks. In our experiments, we perform adaptation on the dev set and report results on the test set. Therefore, the concern that “shifting the dev set may lead to performance degradation when applied to a new, unseen dev split” should not apply.
> - Response to Weakness 4:
> > We have already included DoReMi in all our baseline experiments, so the point about it missing as a baseline is incorrect. We found DoReMi to be a very strong baseline even without downstream task data. In our baseline experiments, we did not consider tDRO, which optimizes the upper bound for all groups and updates both the reference and proxy models to improve the proxy model.
> - Response to Weakness 5:
> > We compare wall-clock GPU hours in Fig. 8, using sentence-transformers embedding data, since that is the largest dataset we used in our experiments.
> - Response to Question 1:
> > We report comparisons on different individual dev splits of BEIR in Tables 6 and 7 in Appendix, showing sensitivity to different dev splits, and demonstrating overall better gains than most baselines.
> - Response to Question 2:
> > We mention this as a crucial requirement for Inf-DDS and many other baselines like DoGE, CRISP, and MultiDDS, which require a dev set for domain adaptation. Assuming you have a dev set that closely resembles the test set, task-adaptive approaches have shown improved performance in many cases. The issue of dev set leakage with the test set, and how to mitigate it during domain adaptation, requires more discussion, which we believe is beyond the scope of this paper.
> - Response to Question 3:
> > We already report this in Figure 8 of the main paper.
>
> We look forward to more suggestions from the reviewer and any other clarification questions that we can answer.

---

> ### Comment · Reviewer_66zW · 2025-11-25
>
> Dear Authors,
>
> Thank you for your responses to the other reviewers so far.
>
> As the discussion period only remains *one week*, for completeness, I would also appreciate your feedback on my concerns, so that the discussion can be fully informed.
>
> I also welcome insights from the other reviewers.
>
> Thanks.

---

> > ### Author Response · Authors · 2025-11-25
> > **Response to Reviewer 66zW**
> >
> > Dear Reviewer 66zW,
> >
> > Thank you for your patience. We have posted our response to the review thread. It appears the earlier version did not grant you read permissions; this has now been corrected. Please let us know if you still cannot access the response or if you have any further questions.
> >
> > Best, Authors.

---

> > > ### Comment · Reviewer_66zW · 2025-11-25
> > >
> > > ## Further Questions
> > >
> > > Thanks for your prompt reply. My concerns still exist.
> > >
> > > 1. **About Weakness 1 & Question 2: Potential Overfitting and Data Leakage (Key Concern)**:
> > > > We emphasize the reliability of the dev set as a key requirement for our work (section 3.1). This is also the case in many previous works like MultiDDS, DoGE, CRISP, etc (lines 125-129). All of these, including ours, assume a reasonable dev set that approximates the test set and has no leakage. Such a dev set can serve as a reliable proxy, which holds true for the widely known BEIR datasets as well as MLDR for multilingual tasks. In our experiments, we perform adaptation on the dev set and report results on the test set. Therefore, the concern that “shifting the dev set may lead to performance degradation when applied to a new, unseen dev split” should not apply.
> > > >
> > > > We mention this as a crucial requirement for Inf-DDS and many other baselines like DoGE, CRISP, and MultiDDS, which require a dev set for domain adaptation. Assuming you have a dev set that closely resembles the test set, task-adaptive approaches have shown improved performance in many cases. The issue of dev set leakage with the test set, and how to mitigate it during domain adaptation, requires more discussion, which we believe is beyond the scope of this paper.
> > >
> > > Yes, I acknowledge the previous works. However, I still believe we should NOT use a portion of the **downstream** test data as the development set to **guide training data sampling**. This causes downstream information **leaking to the training stage**, then overfitting to certain downstream tasks inevitably. This is not beyond the scope of this paper. This is about the key motivation of this work, which is exactly within the scope of this paper.
> > >
> > >
> > > 2. **About Weakness 2: Insufficient Theoretical Analysis**:
> > >
> > > Although I didn't receive your author response about *Weekness 2*,  here I also want to emphasize that the theoretical analysis of a new data sampling redistribution algorithm is very important. For example, the DRO series works have mathematical convergence proofs, which guide clear optimization behaviors.
> > >
> > > As for this paper, my concerns are: How would the data distribution change when the updating steps are small or large? Will the optimization converge or not? Does the *proposed influence* have potential biases and variance?
> > >
> > >
> > > 3. **About Weakness 3 & Question 5: Unclear Generalization Across Domains & Data Anomaly**:
> > >
> > > I didn't receive your response about this. Could you please give some discussions?
> > >
> > >
> > > 4. **About Weakness 4 & Question 4: Baselines**:
> > >
> > > >  We have already included DoReMi in all our baseline experiments, so the point about it missing as a baseline is incorrect. We found DoReMi to be a very strong baseline even without downstream task data. In our baseline experiments, we did not consider tDRO, which optimizes the upper bound for all groups and updates both the reference and proxy models to improve the proxy model.
> > >
> > > Yes, battling against the domain shifts could be done very well even without downstream info leaks.
> > >
> > > And could you clarify why tDRO was excluded from the comparison or discussion? In case I'm not missing something, it is a published work on AAAI, trying to optimize the same data sampling issue in the same area of dense embedding training.
> > >
> > >
> > > 5. **About Weakness 5 & Question 3: Computational Efficiency**:
> > >
> > > > We compare wall-clock GPU hours in Fig. 8, using sentence-transformers embedding data, since that is the largest dataset we used in our experiments.
> > >
> > > Yes, thanks for the response. Could you please give more detailed computational cost breakdowns? This will make your clarification more thorough.
> > >
> > >
> > > 6. **About Question 1: Dev Split Sensitivity**:
> > >
> > > > We report comparisons on different individual dev splits of BEIR in Tables 6 and 7 in Appendix, showing sensitivity to different dev splits, and demonstrating overall better gains than most baselines.
> > >
> > > Tables 6&7 only compare different optimization methods with different hyperparameters. Sensitivity of the method to different dev splits is still not discussed. For example, if we *change* some dev splits, especially choosing *completely OOD splits*, how could the final performance change? Will the advantage of your method still hold?

---

> ### Author Response · Authors · 2025-11-27
> **Response to Reviewer 66zw**
>
> > I still believe we should NOT use a portion of the downstream test data as the development set to guide training data sampling
>
> To clarify: we are **NOT** using the test set, nor a subset of it, as a development set. We use a predefined development set provided by the original data authors. If training domains overlap with the target domains, a development set can alternatively be created by withholding a portion of the training data. Importantly, we do not train on the development set; we use it only to approximate rewards for updating the scorer parameters and not the main model parameters in the bi-level optimization setting. If there is concern about data leakage, there are two ways to test:
> 1. dev/test domains are present in the train domains: A simple soln. would be to perform k-fold cross-validation that includes the dev set and check whether including it in training improves performance on the test set.
> 2. dev/test domains are not present in the train domains: There is no straightforward method that we know of to detect leakage beyond standard preprocessing checks, such as looking for duplicates or token overlaps.
>
> We welcome any suggestions from the reviewer for us to test for such leakage.
>
> > my concerns are: How would the data distribution change when the updating steps are small or large?
>
> We initially experimented with our BEIR setup, using FEVER/FiQA as development sets, and observed that the update steps $l$ do have an impact on performance. Using too many steps does not always lead to the best performance. Here $l=0$ row corresponds to static sampling. Reported numbers are NDCG@10.
>
> | Update steps $l$ | FEVER | FiQA |
> |:----------------:|:-----:|:----:|
> |         0        |  57.3 | 29.0 |
> |         1        |  59.4 | 30.9 |
> |         3        |  64.9 | 33.7 |
> |         5        |  62.4 | 33.4 |
> |        10        |  62.9 | 31.2 |
> |        20        |  60.6 | 31.4 |
>
> > Unclear Generalization Across Domains & Data Anomaly
>
> We do not have a clear explanation for this anomaly, but our current hypothesis is that the model may be using Swahili as a kind of proxy dataset to avoid overfitting on the others. It is possible that the model has inferred that training on Swahili does not affect performance on the target languages, since Swahili is not part of the target set and is largely unrelated to most of the other languages.
>
> > And could you clarify why tDRO was excluded from the comparison or discussion? In case I'm not missing something, it is a published work on AAAI, trying to optimize the same data sampling issue in the same area of dense embedding training.
>
> Thank you for pointing this out. We did not include tDRO as a baseline because we were not aware of this paper at the time of writing. Moreover, tDRO only compares with uniform sampling, and it is unknown if it is better than the wide range of more popular and perhaps stronger baselines that we have compared against, such as DoReMi, MultiDDS, CRISP, DoGE, and others.
>
> > Could you please give more detailed computational cost breakdowns? This will make your clarification more thorough.
>
> Time spent (GPU hrs) on training end-to-end versus scorer updates: MLDR updates take longer because the dev set is composed of MLDR documents, which are all very long. In contrast, the Sent-Trans datasets use BEIR, whose documents are relatively short. The MLDR training data contains both long and short documents, whereas Sent-Trans training is limited to a maximum sequence length of 256 tokens.
>
> |   Setting  | Training | Updates |
> |:----------:|:--------:|:-------:|
> | Sent-Trans |   1436   |   178   |
> |    MLDR    |   1324   |   408   |
>
> > Tables 6&7 only compare different optimization methods with different hyperparameters. Sensitivity of the method to different dev splits is still not discussed. For example, if we change some dev splits, especially choosing completely OOD splits, how could the final performance change? Will the advantage of your method still hold?
>
> We want to kindly clarify that Table 7 shows results using different development set splits, not different hyperparameters. DBPedia and Quora are OOD dev splits, so these results already address the concern. For BEIR, most methods, including ours, do not improve on these OOD splits.
>
> Also, if the concern is about using OOD dev data, our sentence transformer setup already does this. We train on the sentence transformer datasets and tune on the BEIR datasets, so there is no direct overlap between the training data and the development or test data. In this case, the model was able to improve upon static sampling as seen in Table 9.

---

### Official Review · Reviewer_knDf · 2025-11-01

**Soundness:** 4
**Presentation:** 3
**Contribution:** 3
**Rating:** 8
**Confidence:** 4

**Summary:**

This paper introduces Inf-DDS, a reinforcement learning–based sampling framework designed to optimize training data selection for general-purpose open-domain dense retrieval systems. Unlike conventional sampling methods that rely on uniform distribution, proportional instance counts, or expert supervision, Inf-DDS adaptively reweights datasets using influence-based reward signals to enhance model performance on a target development set. The approach requires less GPU usage than gradient-based alternatives. Empirical evaluations across various text retrieval tasks show that Inf-DDS improves retrieval performance, including gains in NDCG@10 scores for both multilingual and sentence-transformer models.

This is a well written paper making a meaningful contribution to training dense retrievers using a more principled way to sample datasets.

**Strengths:**

- This work addresses the important topic of sampling datasets and tasks for training general-purpose dense retrievers in a more principled manner.
- The proposed sampling method was shown to improve retrieval performance while requiring considerably less compute over the baselines across multiple datasets.
- The presentation is clear overall, though Fig 2 (step 3) is not consistent with the pseudocode.

**Weaknesses:**

- The choice for the initial dataset sampling probability distribution is not justified.
- The performance across the dev sets is considered to be equally important, while it may not be the case.

**Questions:**

- The paper stresses that larger dataset size doesn’t necessarily translate to more effective training, yet the dataset sampling probability distribution is initialized proportional to the dataset size. Why?
- The results shown in Fig 4b is unintuitive. For test domains for which training data is available (FEVER and HotpotQA), why aren’t the matching training domains given the highest weight? Is it just Inf-DDS being imperfect?

---

> ### Author Response · Authors · 2025-11-20
> **Response to Reviewer knDf**
>
> We thank the reviewer for their valuable comments and suggestions for our work and look forward to an engaging discussion.
>
> - Response to the suggestion in Strength 3:
> > Thank you for pointing it out. We wanted to indicate an increase in influence in our algorithm (maximization objective), but since we are using a decrease in loss (minimization) as our reward, we have negated it in the figure. We will make this more explicit in the main paper.
> - Response to Weakness 1:
> > For BEIR, we experiment with various initializations. For sentence transformers and the multilingual setting, we start with the sampling distribution that led to matching numbers from the original works.
> - Response to Question 1:
> > Initializing with proportional sampling always or uniform sampling always does not always give the best performance. In our sentence transformer experiments, we found that uniform initialization is better. Although S2ORC, PAQ, and WikiAnswers together make up over 50 percent of the sampling mass under proportional sampling, they tend to perform worse than uniform initialization. This is because MSMarco, which is a relatively small dataset, is a very strong retrieval training dataset.
> - Response to Question 2:
> > We cannot confidently say which dataset should be upsampled, but we believe this can be related to the size and quality of the MSMarco dataset, which generally tends to give good performance across domains. For example, in Fig. 4a first column, you can see MSMarco overall correlates highly with overall performance across different domains.

---

> > ### Author Response · Authors · 2025-11-27
> > **Response to reviewers**
> >
> > Hi,
> > Thank you for taking the time to review our paper. We look forward to your feedback and any comments you may have on our responses.
> >
> > Regards, Authors.

---

### Official Review · Reviewer_9xTa · 2025-11-01

**Soundness:** 2
**Presentation:** 3
**Contribution:** 2
**Rating:** 4
**Confidence:** 3

**Summary:**

This paper proposes Inf-DDS, a dynamic data sampling method for domain-adaptive text retriever training.

**Strengths:**

The problem of selecting informative samples during training is important, especially for reducing computation or improving convergence.
The core idea of prioritizing domains by observed dev improvement (rather than gradient alignment) is intuitive and close to the end goal.

**Weaknesses:**

1. The paper optimizes proxy losses (e.g., InfoNCE / KD loss deltas) but does not demonstrate that these correlate with ranking metrics such as NDCG@10.
2. Influence estimation requires extra forward/backward or Hessian-vector steps. The paper calls the method efficient but does not report GPU-hours / wall-clock / memory, so it is unclear whether gains outweigh the additional compute.
3. The paper alternates between linear-normalized influence weights and softmax Reptile updates, but does not clearly state which variant is used where, nor how τ affects training, hindering reproducibility.
4. No significance testing, so stability and generalizability remain unclear.

**Questions:**

1. For BEIR / Sent-Trans / MLDR, could the authors clearly specify:
(a) the training loss used for the retriever,
(b) the proxy loss used to estimate influence,
(c) the metric M whose change is used to define influence?

2. The paper alternates between describing:
linear-normalized influence weighting, and softmax weighted Reptile updates. So which update rule is actually used in the main reported results？

---

> ### Author Response · Authors · 2025-11-20
> **Response to Reviewer 9xTa**
>
> We thank the reviewer for their valuable comments and suggestions and look forward to an engaging discussion.
>
> - Response to Weakness 1:
> > Previous work by Xu et al., 2024 (https://www.arxiv.org/pdf/2402.06216) has shown equivalence between minimization of InfoNCE and metrics like NDCG@10 and MRR.
> - Response to Weakness 2:
> > Exact GPU hours are difficult to compute when using dynamic sampling, since it depends on the maximum sequence length of the samples in the dataset, which varies. We report approximate wall-clock time in Figure 8, which shows overall lower time because we perform just one training run compared with DoGE or DoReMi.
> - Response to Weakness 3:
> > We use τ=0.02 to scale softmax scores during retrieval loss, which is generally a well-known constant used in the range of 0.01 to 0.02 (ref: https://arxiv.org/abs/2308.03281, https://aclanthology.org/2024.acl-long.642/). Since we have 2 sets of updates: one for the scorer that models the sampling distribution, which uses normalized influence weights, and one for reusing the gradients for the reptile update of the model. Both of these are used simultaneously during Inf-DDS updates.
> - Response to Weakness 4:
> > We apologize for not adding statistical tests for performance comparison. We will add paired tests for comparison.
> - Response to Question 1:
> > We mention this in our experimental setup in Lines 281, 290, and 300 and in more detail in Appendix A. In our setting a), b), and c) are the same in all cases.
> - Response to Question 2:
> > We answer this in response to Weakness 3.

---

> > ### Author Response · Authors · 2025-11-27
> > **Response to reviewers**
> >
> > Hi,
> > Thank you for taking the time to review our paper. We look forward to your feedback and any comments you may have on our responses.
> >
> > Regards, Authors.

---

### Official Review · Reviewer_j7wf · 2025-11-11

**Soundness:** 3
**Presentation:** 3
**Contribution:** 2
**Rating:** 4
**Confidence:** 4

**Summary:**

This paper introduces Inf-DDS, a reinforcement-learning–based framework for domain adaptation in text retrievers. Instead of relying on static or gradient-based sampling, Inf-DDS uses influence scores—measuring how each training dataset affects performance on development sets—to update a sampling policy iteratively. It employs online proxy models and Reptile-style meta-updates to efficiently reuse gradients, thereby reducing GPU overhead.

Experiments on BEIR, MLDR, and Sentence-Transformers training corpora show that Inf-DDS consistently improves NDCG@10 over MultiDDS, DoReMi, DoGE, and static sampling, with gains of up to +5.03 on MLDR and +0.94 over expert-curated weights on MiniLM. The method produces more stable sampling trajectories, generalizes across heterogeneous domains, and reduces compute by 1.5×–4× compared to gradient-based baselines.

**Strengths:**

* Novel influence-based reward mechanism offering more stable, interpretable sampling than gradient-based baselines.
* Computational efficiency via gradient reuse and partial subsampling.
* Clear motivation bridging influence functions with adaptive sampling.

**Weaknesses:**

* Reward estimation cost: computing per-domain influence still scales poorly for very large dataset pools; proxy reliance may not generalize.
* Influence score stability: although more stable than gradient-based methods, influence estimation still depends on the correctness of the proxy update steps.
* Initialization sensitivity: Inf-DDS performance depends heavily on the initial sampling distribution.
* Overfitting to dev sets: using dev-based rewards risks domain leakage.
* Influence effect on unexpected domains (e.g., Swahili in MLDR) remains unexplained and raises questions.

**Questions:**

Influence computation robustness: How does the method behave when proxy model updates and target metric diverge? Can influence computation amplify noise under distribution shifts?

Granularity question: Could instance-level or cluster-level influence scoring outperform dataset-level scoring? What prevents InfDDS from integrating finer-grained sampling within each dataset?

---

> ### Author Response · Authors · 2025-11-20
> **Response to Reviewer j7wf**
>
> We thank the reviewer for their valuable comments and suggestions for our work and look forward to an engaging discussion.
>
> - Response to Weakness 1:
> > In our experiments, we scale up to Sentence-Transformers training pair data, which is a well known and very large retrieval dataset consisting of more than 440 million training pairs across 33 datasets.
> - Response to Weaknesses 3 & 4:
> > Both weaknesses mentioned by the reviewer are already listed in our paper as limitations, and they are also limitations for many prior works. Domain leakage risk exists if the development set overlaps with the test set, but we believe a good development set that matches the test set distribution can give a good approximation for rewards. This assumption also holds for many baselines we compared with, including MultiDDS, DoGE, and CRISP.
> - Response to Question 1:
> > Our proxy model updates (InfoNCE) and target metric (NDCG@10) are shown to be equivalent by Xu et al., 2024 (https://www.arxiv.org/pdf/2402.06216). We are unsure what the reviewer means by “diverging metric” and “amplification of noise under distribution shifts.” It would be helpful if the reviewer could clarify the question so that we can answer it more precisely. From our understanding of the question, noise will be amplified if the approximations for influence are poor (for example, due to pretraining bias, undertraining on some domains, or disparate influence scores when the target metric is easier for some domains). For instance, for languages with high fertility, sequence length differences can cause a distribution shift.
> - Response to Question 2:
> > Finer level granularity approaches can outperform dataset level approaches, but many implementation caveats can hinder their performance. First, CRISP requires maintaining a large number of clusters during training, which can be implemented via scalarization or sampling. Sampling may require keeping many open files or datasets during training, which we found to be prone to memory and OS related errors. Second, scalarization creates a single data source with per-instance scores to scale the loss term; when scaling this to all 440M samples from Sentence-Transformers data, storing these instance-level sampling probabilities can lead to precision issues, such as low gradient magnitude and diminishing loss terms. We did not find a reliable workaround to mitigate this in our experiments. We believe dataset-level approaches are simpler to train when samples within a dataset are relatively homogeneous. For Inf-DDS, storing fine-grained sampling probabilities would face similar caveats and additional issues, such as how to renormalize influence among samples without knowing how to compute a global normalization constant Z. For reference, dataset-level optimizations have worked well for retrieval when instructions are added (https://aclanthology.org/2023.findings-acl.71/, https://aclanthology.org/2023.findings-acl.225/ ).

---

> > ### Author Response · Authors · 2025-11-27
> > **Response to reviewers**
> >
> > Hi,
> > Thank you for taking the time to review our paper. We look forward to your feedback and any comments you may have on our responses.
> >
> > Regards, Authors.

---

### Meta-Review · Area_Chair_S9V9 · 2026-01-07

**Summary:**

This paper introduces influence scores in an RL framework to help find an improved sample strategy for diverse corpora/tasks, and thus is closely related to a range of data-centric methods that have recently emerged as a strategy for improving model performance. The authors study this in the context of embedding models for Text Retrieval.

As an area chair reviewing submissions with an average rating below a reasonable threshold for acceptance (typically top 25%), I tend to look for signals that at least a subset of reviewers strongly defend/argue in support of the submission (suggesting that the paper would appeal to a sub-community, if not to everyone). For this particular submission with a low average rating, Reviewer knDf's positive score would potentially indicate such a scenario. Upon closer look, however, the review is fairly minimal, and no clear justification for the high score is clear from the Review (e.g., why is this an 8 and not a 6). On balance, I thus regret that this submission is, in its current state, below the acceptance threshold for ICLR.

**Reviewer Concerns:**

The authors admit that substantial concerns still remain (e.g. "Both weaknesses mentioned by the reviewer are already listed in our paper as limitations, and they are also limitations for many prior works") or make promises of including results without providing them in the rebuttal ("We apologize for not adding statistical tests for performance comparison. We will add paired tests for comparison."). Thus, several concerns still remain.

**Reviewer Scores:**

It seems unlikely that substantial reviewer score adjustments would have resulted from the rebuttal process due to the nature of the rebuttal not containing a substantial amount of new empirical evidence or methodological insight. The exception to this is the slightly more detailed response to Reviewer 66zW, but even if that Reviewer increased their score, it is unlikely this would reach the realistic bar for acceptance.

---

### Decision · Program_Chairs · 2026-01-26

Reject